# CG-BENCH: CLUE-GROUNDED QUESTION ANSWERING BENCHMARK FOR LONG VIDEO UNDERSTANDING

**Guo Chen**[1,*]**, Yicheng Liu**[1,*]**, Yifei Huang**[2,3,*]**, Yuping He**[1]**, Baoqi Pei**[2,4]**, Jilan Xu**[2,5]
**Yali Wang**[2]**, Tong Lu**[1]**, Limin Wang**[1,2,†]
[1]State Key Laboratory for Novel Software Technology, Nanjing University
[2]Shanghai Artificial Intelligence Laboratory [3]The University of Tokyo
[4]Zhejiang University, [5]Fudan University
`chenguo1177@gmail.com`

## ABSTRACT

The existing video understanding benchmarks for multimodal large language models (MLLMs) mainly focus on short videos. The few benchmarks for long video understanding often rely on multiple-choice questions (MCQs). Due to the limitations of MCQ evaluations and the advanced reasoning abilities of MLLMs, models can often answer correctly by combining short video insights with elimination, without truly understanding the content. To bridge this gap, we introduce CG-Bench, a benchmark for clue-grounded question answering in long videos. CG-Bench emphasizes the model's ability to retrieve relevant clues, enhancing evaluation credibility. It includes 1,219 manually curated videos organized into 14 primary, 171 secondary, and 638 tertiary categories, making it the largest benchmark for long video analysis. The dataset features 12,129 QA pairs in three question types: perception, reasoning, and hallucination. To address the limitations of MCQ-based evaluation, we develop two novel clue-based evaluation methods: clue-grounded white box and black box evaluations, assessing whether models generate answers based on accurate video understanding. We evaluate multiple closed-source and open-source MLLMs on CG-Bench. The results show that current models struggle significantly with long videos compared to short ones, and there is a notable gap between open-source and commercial models. We hope CG-Bench will drive the development of more reliable and capable MLLMs for long video comprehension. All annotations and video data are available at `https://cg-bench.github.io/leaderboard/`.

## 1 INTRODUCTION

Recently, video understanding has made significant progress with the advent of multimodal large language models (MLLMs). To evaluate these models, many recent efforts have been made to create video understanding benchmarks (Li et al., 2023b; Mangalam et al., 2024; Liu et al., 2024e), providing assessments of model comprehension capabilities and clues for future improvement.

Since early benchmarks only focus on short video clips, recent works have started to create benchmarks (Fu et al., 2024a; Wu et al., 2024b; Zhou et al., 2024; Huang et al., 2024) for longer videos ($\geq$ 10 minutes). However, these works employ multiple-choice questions (MCQ), where the difficulty level is heavily influenced by the configuration of negative options. In such scenarios, models (Chen et al., 2023d; Li et al., 2024a; Zhang et al., 2024b; Lin et al., 2024) tend to focus on only general video knowledge and use elimination to avoid selecting the negative options. As a result, the models can achieve correct answers without genuinely engaging with the relevant video content, leading to a lack of trustworthiness. One illustration can be found in question 2 of Figure 1, the option 'A' can be easily eliminated based purely on textual information. Recently, the NExT-GQA (Xiao et al., 2024) benchmark tries to address the problem of credible models by incorporating temporal grounding into MCQ. However, NExT-GQA is limited to the NextQA (Xiao et al., 2021) dataset, which lacks di-

---

*Equal contribution. [†] Corresponding author.

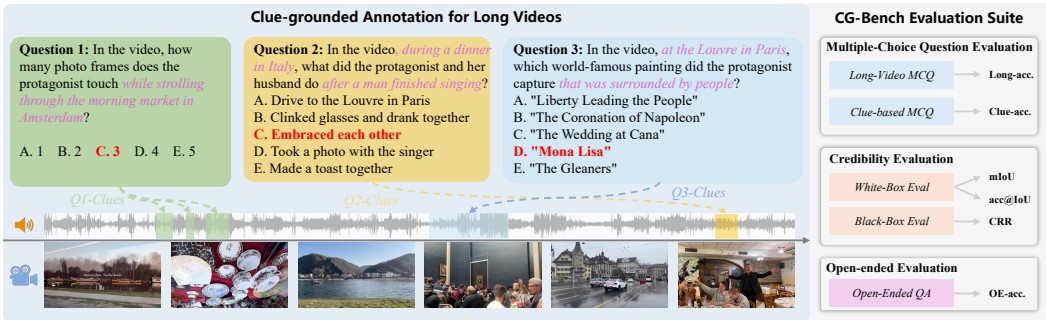

Figure 1: *Left:* examples of CG-Bench's clue-grounded annotation. To correctly answer the questions, models need to ground their reasoning into the correct clue. *Right:* CG-Bench provides an evaluation suite with two novel credibility evaluation criteria while supporting both MCQ and open-ended evaluations.

versity and primarily consists of short videos. A comprehensive benchmark for credibly evaluating *generalist* MLLMs for long video understanding, is still missing in the research community.

To make up this gap, we introduce **CG-Bench**, illustrated in Figure 1, a novel benchmark designed to evaluate clue-grounded question answering in long videos. In contrast to traditional benchmarks that focus primarily on the accuracy of question answering, **CG-Bench** goes a step further by evaluating whether the model bases its answers on relevant clues within the video. **CG-Bench** designs two novel clue-based evaluation methods to provide more reliable model performance assessments. 1) *clue-grouded white box evaluation* requires the model to directly provide the clue interval corresponding to the question while selecting the correct answer. 2) *clue-grouded black box evaluation* requires the model to align the accuracy of video-level and clue-level MCQ. Furthermore, we propose a novel heuristic method, aided by human-annotated clues, for open-ended QA evaluation, to effectively balance the cost and performance.

CG-Bench features 1,219 meticulously curated videos and 12,129 human-annotated question-answer-clue (QAC) triplets, establishing it as the largest and held-out VideoQA and question grounding benchmark for long videos. It employs a highly detailed manual classification system, organizing each video into 14 primary categories, 171 secondary categories, and 638 tertiary categories. The benchmark includes three main question types: perception, reasoning, and hallucination. Perception questions are further divided into 10 subcategories, such as object and attribute recognition, while reasoning questions are categorized into 12 subcategories, including relation reasoning, etc.

We evaluate a range of closed-source and open-source MLLMs using this benchmark. The commercial models, GPT-4o (OpenAI, 2024) and Gemini-1.5 Pro (Anil et al., 2023) achieve scores of 53.9 and 43.4, respectively, with 128 frames for long-video multiple-choice questions. The leading open-source MLLM, Qwen2-VL-72B (Wang et al., 2024b), scores 51.4 under the same conditions, indicating its initial benchmarking against GPT-4o. However, our credibility assessments and open-ended evaluations reveal a significant drop in accuracy for existing MLLMs, with scores decreasing from 53.9 to 21.7. This underscores the considerable room for improvement in current MLLMs for long video understanding. We hope this benchmark can become a vital tool for advancing research and development of more reliable and capable MLLMs.

## 2  RELATED WORK

**Multimodal Large Language Models (MLLMs)** have rapidly gained popularity due to their proficiency in integrating visual and textual information (Liu et al., 2024a; 2023; Chen et al., 2023d; Wang et al., 2022; 2024d). Recent advancements, such as LLaVA-Next-Video (Zhang et al., 2024b), LLaVA-OneVision (Li et al., 2024a), InternVL2 (Chen et al., 2024e) and Eagle-2 (Li et al., 2025), focus on enhancing MLLMs by integrating LLM backbones with visual encoders and specialized adapters, or creating higher-quality multimodal instruction data. This results in improved performance across tasks that involve both text and images.

Another area of focus is multimodal video understanding. Most models (Chen et al., 2024e; Li et al., 2023a; Maaz et al., 2023; Pei et al., 2024; Huang et al., 2018; 2020b) are optimized for short videos, typically a few seconds or at most a few minutes, without exploring their visual understanding with

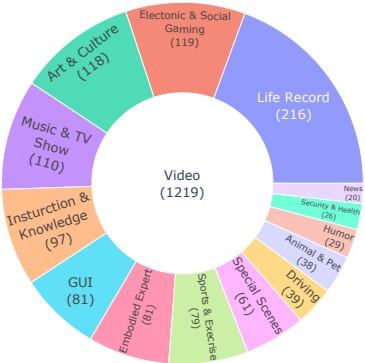 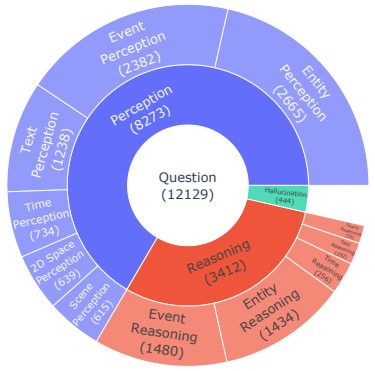

Figure 2: Distribution of video root categories, displaying the number of videos within each category.

Figure 3: Distribution of question root types, illustrating the frequency of different question types.

longer context. In response, researchers have explored methods such as compressing video frames into fewer visual tokens to allow for the handling of longer videos, as seen in models like LLaMA-Vid (Li et al., 2023c), MovieChat (Song et al., 2024), MA-LMM (He et al., 2024), VideoChat-Flash (Li et al., 2024b) and Oryx (Liu et al., 2024f). In addition, LongVA (Zhang et al., 2024a) and LongViLA (Xue et al., 2024) explore the system-level optimization for long-context MLLMs which can natively support long video understanding. Despite the continuous proposal of various MLLMs, their real-world performance in long video understanding is still under explored.

**MLLM Benchmarks.** The development of benchmarks is becoming increasingly essential, especially for evaluating the MLLM performance in video understanding tasks. As the field develops, various benchmarks have been established to assess MLLMs across different modalities and video lengths. Previous efforts primarily focused on short videos, with traditional specialized VideoQA datasets like TVQA (Lei et al., 2018), NextQA (Xiao et al., 2021), and benchmarks for MLLM like VideoBench (Ning et al., 2023), MVBench (Li et al., 2023b) and EgoSchema (Mangalam et al., 2024). MVBench provides a comprehensive framework for evaluating general temporal understanding capabilities through question-answering on short clips, while EgoSchema focuses on egocentric video understanding with multi-choice questions. The videos in these benchmarks typically range from a few seconds to several tens of seconds, making them similar to image benchmarks and thus hindering the development of general video LLMs.

Recently, several works such as VideoMME (Fu et al., 2024a), CinePile (Rawal et al., 2024), MLVU (Zhou et al., 2024), LongVideoBench (Wu et al., 2024b), MoVQA (Zhang et al., 2023b), HourVideo (Chandrasegaran et al., 2024), and LVBench (Wang et al., 2024c), have introduced long video benchmarks to evaluate MLLMs. VideoMME constructs a diverse video MCQ dataset, incorporating multimodal evaluations with visuals, subtitles, and audio. MLVU designs a range of tasks that focus on granular detail understanding to assess long video comprehension capabilities. However, a common limitation of these benchmarks is their reliance on MCQs, where the difficulty is heavily influenced by the construction of negative options. This allows MLLMs to often eliminate incorrect answers using sparse frames and common sense reasoning, which can inflate performances. With our clue interval annotation, CG-Bench enhances the evaluation quality of MLLMs in long video understanding by introducing new evaluation mechanisms on credibility.

## 3 CG-BENCH

### 3.1 DATASET CONSTRUCTION

The dataset construction process of CG-Bench consists of three steps: video collection, question-answering-clue annotation, and quality review iteration. We provide details as follows.

**Video Collection.** To avoid using videos that have been used for pre-training by existing MLLMs, we manually collect videos from the internet and provide new annotations on them. To facilitate the collection of raw videos from the Internet, we define 14 root domains as listed in Figure 2. During the collection process, we manually assign a brief tag (4-8 words) to categorize the content of each video. This supplementary tagging helps to ensure the diversity of the videos. We define

a video to be long if it exceeds 10 minutes in duration. Accordingly, we collected videos longer than 10 minutes while considering the distribution of video duration. Furthermore, we retain the accompanying subtitles and audio to provide multimodal information. We carefully review and filter the videos manually for 7 rounds. More details about the video collection can be found in the supplementary material.

**Question-Answer-Clues Annotation.** After collecting the raw video data, we annotate it with high-quality question-answer-clue (QAC) triplets. To ensure question diversity, we establish a taxonomy with three types: Perception, Reasoning, and Hallucination. As shown in Figure 3, Perception and Reasoning questions are further divided into 10 and 14 subcategories, respectively, while Hallucination questions combine elements of both. Annotators are instructed to include negative options to create a multiple-choice QA format, facilitating straightforward and cost-effective assessments. To minimize expression loss, annotators use their native language during the annotation process. Each video is annotated with 6 to 15 QAC triplets, depending on its duration. To ensure consistency in QAC triplets, we standardized the annotation process by first annotating the QA pairs and then identifying the clues. Annotators must watch the entire video, select a question type from the predefined categories, and then annotate a new question and its corresponding answer. Next, they select one or more intervals from the video to form a QAC triplet. Since the actual clue intervals often consist of multiple short moments, annotating each fragment is costly. Therefore, annotators are required to mark intervals that cover these short moments while ensuring the completeness of each event.

**Review Iteration.** To ensure the difficulty and quality of the dataset, we conduct a repetitive review and iteration process to enhance annotation quality. We reject annotations that do not meet our quality standards and request annotators to revise them. Our quality requirements for annotations and the measures taken to ensure them are as follows: 1) *The rationality of the question, options, and answer*: we conduct manual reviews; 2) *The video dependency of the question, options, and answer*: we input questions and options into GPT-4 and filter out QA pairs that can be answered solely based on pure text; 3) *The difficulty of negative options in multiple-choice questions*: we input the video, questions and options into MLLMs and filter out QA pairs that can be answered using only sparse frames and small models; 4) *The positional diversity of clue intervals:* We monitor the distribution of clue duration and position and provide timely guidance to annotators.

## 3.2 DATASET STATISTICS & COMPARISONS

We present the detailed statistics of our dataset to provide a more comprehensive understanding, including meta-information, QAC triplets, qualitative analysis, and comparison to previous works.

### 3.2.1 DATASET STATISTICS

**Video Meta.** Our dataset comprises a total of 1219 videos with multimodal information, including vision, audio, and subtitles. The duration of the videos varies between 10 and 80 minutes, with a distribution illustrated in Figure 4. Notably, videos that last between 20 and 30 minutes are the most prevalent. This selection process is manual, based on content relevance, which mirrors real-world duration distributions and highlights a long-tail effect for longer videos. As illustrated in Figure 2, each video is classified using a three-tiered tagging system that succinctly encapsulates its content and assigns it to fundamental categories. The primary classification is augmented by a secondary layer of 171 tags and a tertiary layer consisting of 638 tags. This multi-level tagging mechanism guarantees a broad diversity of data content. For a more detailed classification of tags, please consult the supplementary materials.

**QAC Annotation.** CG-Bench includes 12,129 annotations consisting of questions, answers, and clues. Table 1 presents the sentence lengths and totals for the annotated questions and answers, highlighting the linguistic diversity within our dataset. Each QAC triplet is annotated with 4 to 7 negative samples, resulting in an approximately uniform distribution with ratios of options A to H of 12.4%, 14.7%, 12.1%, 14.8%, 15.1%, 16.1%, 11.6%, and 3.1%. There are a total of 14,362 clue intervals across all QAC triplets, with an average duration of 19.24 seconds each. Additionally, we conduct a further analysis of the positions of clue intervals within the video. Figure 5 illustrates the frequency with which each normalized timestamp is represented by intervals. This demonstrates the unbiased nature of our interval annotations and highlights the diversity of our QA content in temporal position.

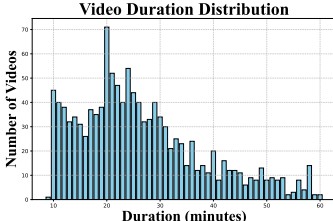

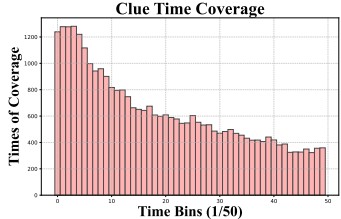

| Annotation Statistics | |
|---|---|
| #QAC Triplets | 12129 |
| #Avg/QAC per video | 9.95 |
| #Avg/Option per QAC | 6.96 |
| #Avg/Clue per QAC | 1.18 |
| #Avg/Words of Questions | 20.07 |
| #Avg/Words of Options | 22.88 |
| #Avg/Duration of Clues | 19.24 |

Figure 4: Video duration distribution, showing the number of videos for different duration intervals.

Figure 5: Clue time coverage, illustrating the frequency of clues across different time bins.

Table 1: Annotation statistics, detailing the number of QAC triplets, questions, options, and clues.

Table 2: Comparison of benchmarks across key aspects: number of videos (#Video), average duration (#Duration), number of QA pairs (#QA Pairs), number of clues (#Clue), annotation method (M/A for manual/automatic), Open-Domain (OD), Open-Ended (OE), Multi-modal (MME), and Credibility (CE) Evaluation.

| Benchmark | #Video | #Dur.(s) | #QA Pairs | #Clue | Anno. | OD | OE | MME | CE |
|---|---|---|---|---|---|---|---|---|---|
| *Question-Clue Grounding* | | | | | | | | | |
| NextGQA (Xiao et al., 2024) | 1,000 | 39.5 | - | 10,531 | M | ✗ | - | - | - |
| Ego4D-NLQ$_{val}$ (Grauman et al., 2022) | 415 | 499.7 | - | 4,554 | M | ✗ | - | - | - |
| Ego4D-NLQ$_{test}$ (Grauman et al., 2022) | 333 | 493.7 | - | 4,005 | M | ✗ | - | - | - |
| MultiHop-EgoQA$_{test}$ (Chen et al., 2024c) | 360 | - | - | 1,080 | A&M | ✗ | - | - | - |
| E.T. Bench$_{test}$ (Liu et al., 2024d) | - | 129.3 | - | 2,011 | M | ✓ | - | - | - |
| RexTime$_{test}$ (Chen et al., 2024a) | - | 141.1 | - | 2,143 | A&M | ✗ | - | - | - |
| **CG-Bench-QG** | 1,219 | 1624.4 | - | 14,362 | M | ✓ | - | - | - |
| *Short-Video QA* | | | | | | | | | |
| TVQA (Lei et al., 2018) | 2,179 | 11.2 | 15,253 | 15,253 | M | ✗ | ✗ | ✗ | ✗ |
| STAR (Wu et al., 2024a) | 914 | 11.9 | 7,098 | 7,098 | A | ✗ | ✗ | ✗ | ✗ |
| NextQA (Xiao et al., 2021) | 1,000 | 44.0 | 8,564 | ✗ | A | ✗ | ✓ | ✗ | ✗ |
| EgoSchema (Mangalam et al., 2024) | 5,063 | 180.0 | 5,063 | ✗ | A&M | ✗ | ✗ | ✗ | ✗ |
| TempCompass (Liu et al., 2024e) | 410 | 11.4 | 7,540 | ✗ | A&M | ✗ | ✗ | ✗ | ✗ |
| RexTime$_{test}$ (Chen et al., 2024a) | - | 141.1 | - | 2,143 | A&M | ✗ | ✗ | ✗ | ✓ |
| MVBench (Li et al., 2023b) | 3,641 | 16.0 | 4,000 | ✗ | A&M | ✗ | ✗ | ✗ | ✗ |
| MMBench-Video (Fang et al., 2024) | 600 | 165.4 | 1,998 | ✗ | M | ✓ | ✓ | ✗ | ✗ |
| **CG-Bench-Clue** | 12,129 | 22.8 | 12,129 | - | M | ✓ | - | ✓ | - |
| *Long-Video QA* | | | | | | | | | |
| EgoTimeQA$_{test}$ (Di & Xie, 2024) | 148 | 492 | 500 | ✗ | A | ✗ | ✗ | ✗ | ✗ |
| MovieChat-1K (Song et al., 2024) | 130 | 500.0 | 1,950 | ✗ | M | ✗ | ✗ | ✗ | ✗ |
| Video-MME (Fu et al., 2024a) | 900 | 1017.9 | 2,700 | ✗ | M | ✓ | ✗ | ✓ | ✗ |
| LongVideoBench (Wu et al., 2024b) | 966 | 1408.0 | 6,678 | ✗ | M | ✓ | ✗ | ✗ | ✗ |
| MLVU (Zhou et al., 2024) | 757 | 720.0 | 2,593 | ✗ | M | ✗ | ✗ | ✗ | ✗ |
| **CG-Bench** | 1,219 | 1624.4 | 12,129 | 14,362 | M | ✓ | ✓ | ✓ | ✓ |

### 3.2.2 COMPARISON WITH PREVIOUS BENCHMARKS

CG-Bench is characterized by its diverse features, allowing it to be compared with three distinct types of benchmarks, as depicted in the three sections of Table 2: Question Clue Grounding, Short-Video QA, and Long-Video QA benchmarks. For the question clue grounding benchmarks, NextGQA (Xiao et al., 2024), Ego4D-NLQ (Grauman et al., 2022), MultiHop-EgoQA (Chen et al., 2024c), E.T. Bench (Liu et al., 2024d), and RexTime (Chen et al., 2024a) are primarily centered around action and egocentric domains. Their videos are sampled from academic datasets. In comparison, the question clue grounding part of CG-Bench, CG-Bench-QG, stands out with the highest number of videos and the longest average length, the diversity of which fosters a broad spectrum of question-grounding queries.

Furthermore, we transform QAC triplets to our novel Short-Video QA benchmark, termed CG-Bench-Clue. When contrasted with prior short video benchmarks such as TempCompass (Liu et al., 2024e), MVBench (Li et al., 2023b) and MMBench-Video (Fang et al., 2024), our CG-Bench-Clue emerges as the ***largest***, ***held-out***, ***open-domain*** and ***multimodal*** Short-Video QA benchmark.

As for the Long-Video QA benchmark, CG-Bench excels in the number of videos, length, quantity of questions, and annotation quality. Owing to our clue interval annotations, CG-Bench further facilitates reliable evaluations for long videos and open-ended evaluations with clue assistance, a feature that sets it apart from existing long video benchmarks like Video-MME (Fu et al., 2024a) and MLVU (Zhou et al., 2024).

### 3.3 EVALUATION

In this section, we describe the evaluation tasks of our CG-Bench which include traditional MCQ, the unique credibility evaluation, and clue-aided open-ended QA evaluation.

#### 3.3.1 MULTIPLE-CHOICE QUESTION EVALUATION

We assess the accuracy of MCQ in two settings: **Long-Video MCQ** and **Clue-based MCQ**. In the Long-Video MCQ setting, the model receives the entire video as input and is required to select the correct answer based on the video, the question, and the candidate options. For the Clue-based MCQ setting, the model is given only the video within the annotated clue interval as input. The model has access only to the clue clip, the question, and the candidate options. It does not have access to the original long video. Since a single QA may correspond to multiple clues, we merge these clues and treat the combined clue as a single, cohesive clue segment.

#### 3.3.2 CREDIBILITY EVALUATION

The ability of a model to identify relevant clues related to questions is a crucial factor in determining its reliability. Therefore, we define a model's reliability based on its proficiency in locating accurate clues when addressing problems. To achieve this, we introduce two clue-grounded mechanisms for credibility assessment: white-box evaluation and black-box evaluation.

**White-Box Evaluation** requires the model to directly output the intervals of clues that can accurately answer the question. This task is similar to video temporal grounding (Lei et al., 2021; Huang et al., 2023). Therefore, we use tIoU (Temporal Intersection over Union) as the evaluation metric. Since each question may correspond to multiple intervals of clues, we allow the model to predict multiple possible intervals. Given a set of prediction $\mathcal{P}$ and ground truths $\mathcal{G}$, the tIoU is defined as:

$$\text{tIoU} = \frac{\sum_{i \in \mathcal{G}, j \in \mathcal{P}} \max(0, \min(b_i, d_j) - \max(a_i, c_j))}{\sum_{i \in \mathcal{G}} (b_i - a_i) + \sum_{j \in \mathcal{P}} (d_j - c_j) - \sum_{i \in \mathcal{G}, j \in \mathcal{P}} \max(0, \min(b_i, d_j) - \max(a_i, c_j))} \times 100\%, \quad (1)$$

where $a_i$, $b_i$ are the start and end timestamps of the $i$-th ground truth interval of $\mathcal{G}$. $c_j$, $d_j$ are the start and end timestamps of the $j$-th predicted interval of $\mathcal{P}$. We calculate the mean IoU (**mIoU**) by averaging the tIoU scores obtained by the model across all question queries. To further improve the robustness of question grounding evaluation, we introduce the **rec.@IoU** metric. This metric measures the probability of successfully recalling clue intervals at various IoU thresholds.

Additionally, we propose **acc.@IoU** that evaluates both MCQ accuracy and clue-grounding ability. A response is considered correct only if the selected answer is accurate and the tIoU exceeds ($>$) a predefined threshold $\tau$. Since locating short-duration clues in the long videos in CG-Bench is inherently challenging, we set the default $\tau$ to be 0 for the more obvious comparison on ablation studies. Setting $\tau = 0$ ensures that acc.@IoU requires the model to select the correct option and produce a time interval that overlaps at least slightly (tIoU $> 0$) with the annotated clue interval, rather than reducing to naive MCQ accuracy. We calculate the **rec.@IoU** and **acc.@IoU** at IoU thresholds of 0.1, 0.2, 0.3, 0.4, and 0.5 to determine the final result.

**Black-Box Evaluation** aims to evaluate the model's ability to seek out clues implicitly. Understanding long videos involves the retrieval of clues distributed across various spatiotemporal locations within the entire video. Therefore, an effective model for long videos should naturally focus on capturing human-annotated clue intervals in its hidden states. However, beyond the explicitly annotated clue intervals, there are likely hidden clues scattered throughout the video that can also help to determine the correct answer. Thus, a model with access to the full video should yield higher accuracy compared to solely relying on the clue interval. In other words, the accuracy of Long-Video MCQ ( **long-acc.**) should be greater than or equal to the accuracy of Clue-based MCQ (**clue-acc.**).

With this insight, for the black box evaluation, we define a new metric called Clue Recovery Rate (**CRR**). This metric evaluates the model's robustness to context dilution, *i.e.*, how stable a model can find related clues from long but diluted video context. CRR is calculated by:

$$\text{CRR} = \frac{\min(\textbf{long-acc.}, \textbf{clue-acc.})}{\textbf{clue-acc.}} \times 100\%, \quad (2)$$

A CRR of less than 100% suggests that the MLLM's ability to retrieve short clues from long video representations is not optimal.

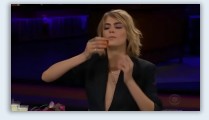 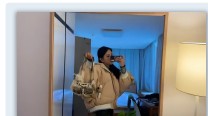

Figure 6: Two examples illustrating the ambiguity challenge of using LLMs for open-ended evaluation. While in different expressions, GT and prediction should both be treated as correct answers.

### 3.3.3 CLUE-AIDED OPEN-ENDED QA EVALUATION

CG-Bench supports open-ended QA evaluation for a more comprehensive assessment. Previous works like MM-Vet (Yu et al., 2023) and MMBench-Video (Fang et al., 2024) have used LLMs to evaluate open-ended QA for images and short videos. However, long videos contain more complex information, leading to ambiguous user-generated questions. This can result in discrepancies between LLM-evaluated scores and the actual QA ability of models, as shown in Figure 6.

To address this, we use a low-hallucination MLLM to evaluate the similarity between text output and visual information. We selected GPT-4o (OpenAI, 2024) as the multimodal evaluator due to its high ranking in benchmarks like OpenCompass (Contributors, 2023) and the Lmsys leaderboard (Chiang et al., 2024), and its relatively low hallucination rate compared to other MLLMs. Since using GPT-4o directly for multimodal judging can still introduce hallucination errors and incur high API costs, we propose a heuristic evaluation method to mitigate biases and reduce costs. First, GPT-4o determines if the output can be evaluated based solely on the text answer. It outputs `yes` or `no`; if not, it requests visual cues by stating, "I need visual clues." This prompts the inclusion of supplementary visual data to aid GPT-4o in its evaluation. By using pre-annotated time intervals with question clues, we sample frames as visual aids, further reducing hallucination errors and costs. We analyze this evaluation method in Sec 4.3, with more details available in the supplementary materials.

## 4 EXPERIMENTS

In this section, we evaluate a wide range of MLLMs using CG-Bench. We first introduce the evaluation setup, followed by quantitative results for both closed-source and open-source models. Finally, we analyze some key factors in the evaluation.

### 4.1 SETTINGS

We first briefly describe the settings used in our experiments. The supplementary material provides more detailed settings.

**Models.** We evaluate the performance of three mainstream commercial models on our CG-Bench: GPT4o (OpenAI, 2024), Gemini-1.5 (Anil et al., 2023), and Claude-3.5, including their different versions. Also, we assess the representative open-source image-MLLMs, such as LLaVA-OV (Li et al., 2024a), Qwen2-VL (Wang et al., 2024b) and InternVL2 (Chen et al., 2024e), video-MLLMs, such as VideoChat2 (Li et al., 2023b).

**Frame Sampling.** For long video understanding, the frame sampling strategy significantly impacts evaluation results. For open-source MLLMs, we make the best use of our computational resources to use as many frames as possible. For closed-source MLLMs, since the local computational resource is no longer a bottleneck, we can use even more frames. We uniformly sample (Wang et al., 2019) 128 frames for Long-video MCQ, and use 32 frames as the for Clue-based MCQ.

**Modality.** We also explore other modalities: subtitles and audio. For subtitles, we employ a uniform sampling method. If the timestamp of a sampled frame falls within the time interval of a subtitle, that subtitle will be included in the analysis. Each subtitle is considered only once to avoid redundancy.

**Prompt.** For MCQ tasks, the model is prompted to provide the uppercase letter corresponding to the correct option. In Open-Ended QA tasks, the model responds freely based on the questions. For the Clue Grounding task, we append the timestamps of each frame and subtitle to enhance the model's time-awareness, requiring it to return nested lists in the format `[[s1, e1], [s2, e2], ...]`. For open-ended evaluation, we require the model to assess the correctness between the predictions and the ground truth and respond with `yes` or `no`.

Table 3: Performance of various open-source and closed-source MLLMs on CG-Bench. We provide human evaluation for showing annotation agreements and the difficulty of our benchmark.

| Models | LLM | #F | | MCQ | | Cred. Eval. | | | | OE |
|---|---|---|---|---|---|---|---|---|---|---|
| | #param | clue | long | clue-acc. | long-acc. | mIoU | rec.@IoU | acc.@IoU | CRR | acc. |
| Random | - | - | - | 14.2 | 14.2 | 0.13 | 0.16 | 0.09 | 100 | 0 |
| Human (full-video) | - | - | - | 92.2 | 90.3 | 35.5 | 51.2 | 29.8 | 97.9 | 83.7 |
| Human (sparse frames) | - | - | 128 | - | 59.9 | - | - | - | - | - |
| GPT4o (text) | - | 0 | 0 | 16.8 | 16.8 | 0.14 | 0.2 | 0.15 | 100 | 2.1 |
| **Open-source MLLMs** | | | | | | | | | | |
| Video-LLAVA (Lin et al., 2023) | 7B | 8 | 8 | 34.2 | 16.2 | 1.13 | 1.96 | 0.59 | 47.4 | 12.3 |
| VideoLLAMA (Zhang et al., 2023a) | 7B | 32 | 32 | 36.8 | 18.4 | 1.21 | 1.87 | 0.84 | 50.0 | 15.8 |
| Videochat2 (Li et al., 2023b) | 7B | 16 | 16 | 35.2 | 19.3 | 1.28 | 1.98 | 0.94 | 54.8 | 18.6 |
| Qwen-VL-Chat (Bai et al., 2023) | 7B | 4 | 4 | 38.3 | 21.6 | 0.89 | 1.19 | 0.42 | 56.4 | 19.4 |
| ST-LLM (Liu et al., 2024c) | 7B | 32 | 64 | 39.6 | 23.8 | 2.23 | 2.86 | 1.13 | 60.1 | 20.7 |
| ShareGPT4Video (Chen et al., 2024b) | 16B | 16 | 16 | 41.4 | 26.7 | 1.85 | 2.65 | 1.01 | 64.5 | 22.0 |
| Chat-UniVi-v1.5 (Jin et al., 2024) | 13B | 32 | 64 | 41.5 | 25.9 | 2.07 | 2.53 | 1.21 | 62.4 | 21.4 |
| ViLA (Lin et al., 2024) | 8B | 14 | 14 | 41.8 | 28.7 | 1.56 | 2.89 | 1.35 | 68.7 | 24.0 |
| GroundVQA (Liu et al., 2024d) | 0.25B | - | 1200 | 27.3 | - | 1.33 | 1.37 | - | - | - |
| GeLM (Chen et al., 2024c) | 7B | - | 100 | - | - | 2.25 | 2.81 | - | - | - |
| ET-Chat (Liu et al., 2024d) | 4B | - | 1fps | 17.6 | - | 1.38 | 1.43 | - | - | - |
| InternVL-Chat-v1.5 (Chen et al., 2023d) | 20B | 10 | 10 | 42.5 | 28.9 | 2.18 | 2.38 | 1.15 | 68.0 | 23.1 |
| MiniCPM-v2.6 (Yao et al., 2024) | 8B | 32 | 32 | 44.6 | 30.1 | 2.35 | 2.61 | 1.04 | 67.5 | 26.6 |
| LongVA (Zhang et al., 2024a) | 7B | 32 | 128 | 42.8 | 28.7 | 2.94 | 3.86 | 1.78 | 67.1 | 25.1 |
| LLaVA-OneVision (Li et al., 2024a) | 7B | 16 | 16 | 43.2 | 31.1 | 1.63 | 1.78 | 1.08 | 72.0 | 25.4 |
| Video-CCAM (Fei et al., 2024) | 14B | 32 | 96 | 43.6 | 29.7 | 2.63 | 3.48 | 1.83 | 68.1 | 25.3 |
| Kangaroo (Liu et al., 2024b) | 8B | 32 | 64 | 45.9 | 30.2 | 2.56 | 2.81 | 1.94 | 65.8 | 24.5 |
| VITA (Fu et al., 2024b) | 8x7B | 32 | 32 | 47.8 | 33.3 | 3.06 | 3.53 | 2.06 | 69.7 | 27.5 |
| Qwen2-VL (Wang et al., 2024b) | 72B | 32 | 128 | 56.2 | 41.3 | 3.58 | 5.32 | 3.31 | 73.5 | 33.6 |
| InternVL2 (Chen et al., 2024e) | 78B | 32 | 32 | 58.5 | 42.2 | 3.91 | 5.05 | 2.64 | 72.1 | 32.5 |
| **Closed-source MLLMs** | | | | | | | | | | |
| GPT-4o-08-06 (OpenAI, 2024) | - | 32 | 128 | 58.3 | 45.2 | 5.62 | 8.30 | 4.38 | 77.5 | 39.5 |
| GPT-4mini-08-06 (OpenAI, 2024) | - | 32 | 128 | 48.3 | 33.4 | 3.75 | 5.18 | 2.21 | 69.2 | 25.4 |
| Gemini-1.5-Pro (Anil et al., 2023) | - | 32 | 128 | 50.1 | 37.2 | 3.95 | 5.81 | 2.53 | 74.3 | 29.3 |
| Gemini-1.5-Flash (Anil et al., 2023) | - | 32 | 128 | 47.0 | 32.3 | 3.67 | 5.44 | 2.45 | 68.7 | 26.3 |
| Claude3.5-Sonnet | - | 32 | 50 | 56.2 | 40.5 | 3.99 | 5.67 | 2.79 | 72.1 | 35.2 |

## 4.2 Main Results

As shown in Table 3, the closed-source MLLM GPT4o (OpenAI, 2024) leads significantly, surpassing other models across all metrics. Notably, GPT4o's long-acc. reaches 45.2%, much higher than Gemini-1.5-Pro (Anil et al., 2023), demonstrating its strong capabilities in long video understanding. Among open-source MLLMs, Qwen2-VL (Wang et al., 2024b) performs impressively, achieving results comparable to GPT4o in long-acc. and clue-acc. Other models underperform due to insufficient context support or inadequate video training. While these MLLMs perform well on MCQ tasks, they experience significant performance drops in credibility and open-ended evaluations on CG-Bench. For instance, GPT-4o's long-acc. falls from 45.2 to 4.38 in Acc@IoU and 39.5 in OE-acc. Additionally, with the same number of sampling frames, GPT-4o achieves a CRR of 77.5, whereas Gemini-1.5-Pro only reaches 74.3, indicating its weaker ability to retrieve short-term clues from long videos. Overall, current MLLMs do not perform well on CG-Bench, suggesting considerable room for improvement in their capability and credibility.

Since it is difficult to input more than 128 frames due to the hardware limitations, we alternatively conducted a human evaluation experiment under constrained visual conditions, to see how severe the "undersampling" issue is for longer video. We uniformly sampled 30 videos from CG-Bench, resulting in 296 questions. For each video, we uniformly sampled 128 frames and asked volunteers to perform an MCQ testing. The resulting accuracy was 59.85% (row 3 in Table 3). This result indicates that our dataset is indeed challenging and that it is difficult to derive solutions from a limited number of frames. It also highlights that even the most advanced models, such as GPT-4o, have ample room for improvement in long video comprehension.

## 4.3 Analysis

Furthermore, we perform a comprehensive analysis of the two leading closed-source MLLMs, GPT4o (OpenAI, 2024) and Gemini-1.5 Pro (Anil et al., 2023), as well as the best performing open-source MLLM, Qwen2-VL (Wang et al., 2024b), on our CG-Bench. In this analysis, we use 1000

Table 4: Impact of different prompts and modalities. Each prompt can be composed of frames (F), frame timestamps (FT), subtitles (S), subtitle timestamps (ST), and audio (A). We conduct the main experiments with GPT4o-0806 (OpenAI, 2024) while studying the audio modality with Gemini-1.5 Pro (Anil et al., 2023).

| model | prompt & modality | clue-acc. | long-acc. | mIoU | Acc@IoU | CRR | OE-acc. |
|---|---|---|---|---|---|---|---|
| GPT4o | S (128 frames) | - | 31.5 | - | - | - | - |
| GPT4o | S (full-video) | - | 34.3 | - | - | - | - |
| GPT4o | F | 65.8 | 51.8 | 3.39 | 10.7 | 78.7 | 35.4 |
| GPT4o | F+FT | $65.3_{(-0.5)}$ | $51.6_{(-0.2)}$ | $5.73_{(+2.34)}$ | $20.4_{(+9.7)}$ | $79.0_{(-0.3)}$ | $36.8_{(+1.4)}$ |
| GPT4o | F+S | $66.7_{(+0.9)}$ | $53.4_{(+1.6)}$ | $3.96_{(+0.57)}$ | $11.2_{(+0.5)}$ | $80.1_{(+1.4)}$ | $38.2_{(+2.8)}$ |
| GPT4o | F+S+ST | $67.1_{(+1.3)}$ | $54.1_{(+2.3)}$ | $5.19_{(+1.80)}$ | $13.2_{(+2.5)}$ | $80.6_{(+1.9)}$ | $38.4_{(+3.0)}$ |
| GPT4o | F+S+FT | $67.4_{(+1.6)}$ | $53.2_{(+1.4)}$ | $7.80_{(+4.41)}$ | $22.3_{(+11.6)}$ | $78.9_{(+0.2)}$ | $37.9_{(+2.5)}$ |
| GPT4o | F+S+ST+FT | $\mathbf{67.5}_{(+1.7)}$ | $\mathbf{54.9}_{(+3.0)}$ | $\mathbf{9.68}_{(+6.29)}$ | $\mathbf{26.7}_{(+16.0)}$ | $\mathbf{81.3}_{(+2.6)}$ | $\mathbf{39.5}_{(+4.1)}$ |
| Gemini | F+S+ST+FT | 62.1 | 45.1 | 9.16 | 20.7 | 72.6 | 23.2 |
| Gemini | F+S+ST+FT+A | $62.3_{(+0.2)}$ | $45.0_{(-0.1)}$ | $9.10_{(-0.06)}$ | $19.8_{(-0.9)}$ | $72.2_{(-0.4)}$ | $23.5_{(+0.3)}$ |

(a) Long Acc    (b) mIoU    (c) Acc@IoU    (d) CRR    (e) Open ended

Figure 7: Impact of sampling frame numbers on different metrics for GPT-4o-0806 (OpenAI, 2024), Gemini-1.5 Pro (Anil et al., 2023) and Qwen2VL-72B (Wang et al., 2024b).

QAC triplets sampled uniformly from all annotations for fast experiments. We report acc.@IoU with $\tau = 0$ for a more obvious comparison.

**Impact of Prompt & Modality.** As shown in Table 4, we conduct the ablation studies on the subset that contains subtitles and explore the impact of different prompts on GPT4o and the effect of the audio modality on Gemini-1.5 Pro. Our findings indicate that all prompt types (FT/S/ST), except video frames (F), provide performance benefits across most metrics. Subtitles contribute more to **long-acc.** than they do to **clue-acc.**. Additionally, the inclusion of timestamp information (FT/ST) is critical for interval prediction. Timestamps from both frames and subtitles enhance IoU-related metrics, revealing a complementary effect. When both FT and ST are added simultaneously, **mIoU** increases from 3.39 to 9.68, and **Acc@IoU** rises from 10.7 to 26.7. When S, FT, and ST are all used in the prompt, the model achieves the best performance across all metrics. In contrast, our exploration of the audio modality (A) revealed that audio does not yield significant performance gain and, in some cases, even slightly degrades the results, as shown in Table 4. Finally, we conduct experiments using only subtitles from 128 frames versus the full video. The results show that while subtitles offer useful semantic cues, their impact is significantly reduced when visual input is included. This suggests that our benchmark favors visual signals.

**Impact of Frame Number.** As illustrated in Figure 7, we conducted experiments to analyze the performance across various metrics as the number of frames increases. Overall, the performance of all three MLLMs gradually improves with the addition of more frames, with GPT-4o consistently outperforming the others across all metrics. For **long-acc.** and **OE acc.**, Qwen2VL achieves performance comparable to GPT-4o. However, compared with Qwen2VL, Gemini excels in terms of mIoU and Acc@IoU. Regarding **CRR**, GPT-4o demonstrates greater consistency between clue-acc. and long-acc. across more frames, indicating its superior reliability in long video understanding. For open-ended QA, Gemini's higher refusal rate results in a noticeable decline in performance.

**Open-ended Evaluation Quality.** To assess the stability and accuracy of various MLLMs as evaluators, we utilized four models—Gemini, Qwen2VL, Claude, and GPT-4o—each of which evaluated GPT-4o's predictions five times. Human evaluations of GPT-4o's predictions are also conducted for reference. The results, shown in Figure 8, indicate that GPT-4o has the highest stability and the smallest deviation from human-assigned scores. Furthermore, Table 5 explores the impact of different evaluation methods. When evaluators were provided only with ground truth (col. "GT") or visual information (col. "Vis"), the scoring bias (absolute difference) between human and model-

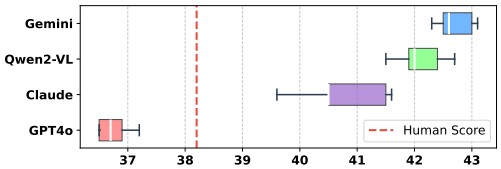

Figure 8: Comparison of using different LLMs as open-ended evaluators for GPT-4o's outputs.

Table 5: Comparison of different modes: GT-only, visual-only, GT+vision and heuristic (Ours).

|  | GT | GT+Vis | Vis | **Ours** |
|---|---|---|---|---|
| Bias(%)↓ | 12.4 | 6.4 | 17.0 | **1.0** |
| Time (s)↓ | **741** | 20,040 | 19,640 | 3,600 |
| Price ($)↓ | **0.05** | 6.1 | 6 | 2 |
| Trigger Rate (%)↓ | 0 | 100 | 100 | **14** |
| Trigger Recall Rate (%)↑ | 0 | 100 | 100 | **88** |

Figure 9: Long-Video-MCQ Accuracy grouped by video duration for GPT4o-0806 with 128 frames.

Table 6: Impact of different frame sampling strategies on long-acc. for GPT4o-0806.

| #Frames | Resolution | Sampling Strategy | long-acc. |
|---|---|---|---|
| 128 | Low | Uniform | **53.9** |
| 50 | Low | Uniform | 46.7 |
| 50 | Low | Keyframe | 45.7 |
| 50 | High | Uniform | 51.0 |

based evaluation increased. While fully leveraging visual information (col. "GT+Vis") improved evaluation accuracy, it also significantly increased the time and cost required. Our proposed heuristic evaluation method achieves the lowest evaluation bias. Additionally, we manually annotated 200 evaluation samples to determine the necessity of visual request triggers. From the bottom block in Table 5, the statistics show that our method achieved a visual request trigger rate (the probability that the model triggers "visual clues required") of 14%. The recall rate of this triggering achieves 88%. This proves that our approach effectively balances cost and performance.

**Performance grouped by Video Duration.** We grouped videos by duration and evaluated the **long-acc.** performance of GPT-4o-0806 using 128 frames. Figure 9 shows that the model struggles with undersampling, especially for longer videos.

**Impact of Frame Sampling Strategy.** We investigate how different frame sampling strategies affect performance. To expedite testing, we primarily evaluated GPT4o-0806 using 50 uniformly sampled frames, focusing on the **long-acc** metric. The experiment consists of three parts: 1) low resolution, 2) high resolution, and 3) keyframe extraction (via FFmpeg) combined with low resolution. As shown in Table 6, higher resolution offers some improvement, while keyframe extraction has no significant impact.

## 5 CONCLUSION AND FUTURE WORK

In this paper, we introduce CG-Bench, a novel benchmark designed to evaluate clue-grounded question answering capabilities in long video understanding. Unlike existing benchmarks that focus on short videos or rely solely on multiple-choice questions, CG-Bench emphasizes the importance of models retrieving and grounding their answers in specific video segments, enhancing evaluation credibility. CG-Bench includes 1,219 manually curated videos organized into a detailed three-tier system, with 12,129 QA pairs covering perception, reasoning, and hallucination question types. This provides a comprehensive and diverse dataset for assessing MLLMs. We propose two clue-based evaluation methods—clue-grounded white-box and black-box evaluations—that offer novel ways to determine whether models genuinely comprehend video content or merely rely on superficial cues. Extensive experiments with various closed-source and open-source MLLMs reveal that current models significantly underperform in long video understanding compared to short videos. We hope that CG-Bench will serve as a valuable resource for the research community, driving the development of more trustworthy and capable MLLMs for long video understanding.

**Acknowledgement.** This work is supported by the National Key R&D Program of China (No. 2022ZD 0160900), the National Natural Science Foundation of China (No. 62372223 and U24A20330), Jiangsu Frontier Technology Research and Development Program (No. BF2024076), Postgraduate Research & Practice Innovation Program of Jiangsu Province (No. KYCX24_0260), and Nanjing University-China Mobile Communications Group Co., Ltd. Joint Institute.

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

# APPENDIX

# A    ANNOTATION

## A.1    QUALITY CONTROL

During the annotation process, we implement a quality control system as illustrated in Figure 10. We use a batch increment method for data iteration, reviewing each batch of about 1,000 items.

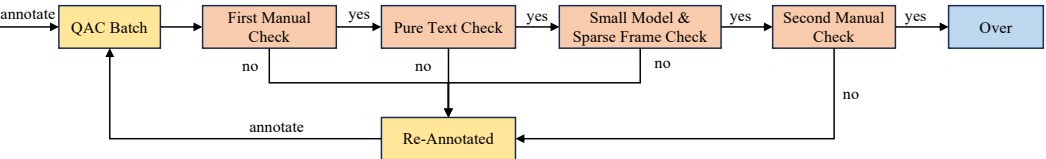

Figure 10: Annotation Quality Control Flowchart.

First, a manual review checks for typos and ensures question quality. We focus on two main aspects: clarity and granularity. Questions must have a clear anchor point, such as an event or scene, to avoid confusion. The granularity should be appropriate; overly broad questions provide too many easy clues, which undermines our goal of testing the model's ability to pinpoint clues.

Next, to ensure question difficulty, we conduct tests using LLM, such as GPT4 (OpenAI, 2023) and Qwen2.5 (Yang et al., 2024a), with pure text and small MLLM, like InternVL2-2B (Chen et al., 2024e) and InternVL2-4B, with sparse frames. The pure text test ensures that questions and options don't reveal too much information, allowing models to answer without visual data.

Finally, the second manual review catches other remaining issues, resulting in the final test set.

We provide two examples of filtered samples for "Small Model & Sparse Frame Check" in Figure 11 and Figure 12. For Figure 11, the protagonist is cycling in a first-person view, and the outfit appears throughout the video. For Figure 12, the climbing wall is a prominent target, and the distinctions between the options are very clear, requiring minimal comprehension.

And here is another example of filtered sample for "Pure Text Check":

***In the video, according to the content shown in the PPT, the teacher talked about ethylene. So what will be produced after ethylene is oxidized by potassium permanganate?***
*A. acetone*
*B. acetic acid*
*C. acetaldehyde*
***D. carbon dioxide***
*E. carbon monoxide*

This QA is essentially a simple chemistry question and therefore did not pass the check.

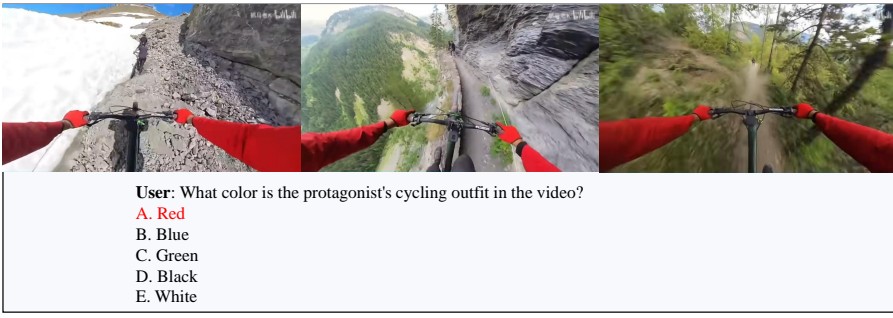

User: What color is the protagonist's cycling outfit in the video?
A. Red
B. Blue
C. Green
D. Black
E. White

Figure 11: Example 1 of filtered QA by "Small Model & Sparse Frame Check".

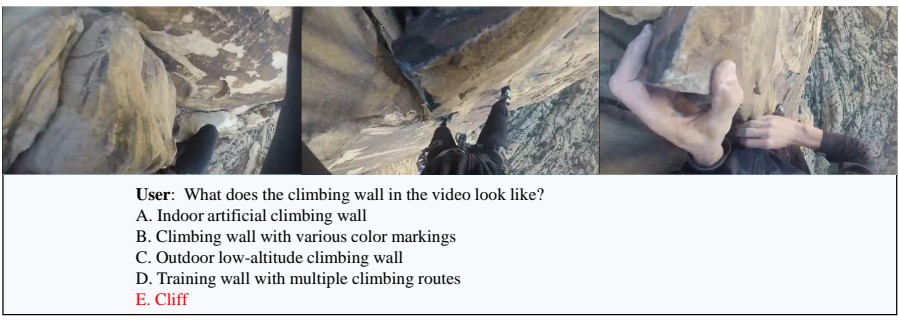

User: What does the climbing wall in the video look like?
A. Indoor artificial climbing wall
B. Climbing wall with various color markings
C. Outdoor low-altitude climbing wall
D. Training wall with multiple climbing routes
E. Cliff

Figure 12: Example 2 of filtered QA by "Small Model & Sparse Frame Check".

## A.2 STATISTICS

**Question Categories and Definition.** We list all question categories in Table 7. We also provide a rough definition of each question type:

- **Entity Recognition**: Identifying entities within the context, focusing on recognizing specific objects or entities present in the scene.
- **Entity Counting**: Addressing the quantity of entities, focusing on counting the number of specific objects.
- **Entity Attribute**: Exploring the attributes of entities, such as shape, color, material, etc.
- **Entity State**: Investigating the state of an object, including its current condition and any changes over time, focusing on the status of the entity and its evolution.
- **Event Recognition**: Identifying events within the context, focusing on recognizing specific occurrences or actions taking place in the scene.
- **Event Counting**: Counting the occurrences of events, focusing on how many times a particular event takes place within the context.
- **Scene Recognition**: Identifying and understanding the scene where an event takes place. Questions may explore details about the setting, such as its characteristics, background elements, or overall atmosphere.
- **Text Recognition**: Identifying the content of text, focusing on recognizing specific text elements within the context.
- **Text Counting**: Addressing the number of specific aspects of the text, focusing on the number of certain elements or directions within the text.
- **Time Localization**: Identifying the temporal range or specific time points of an event.
- **Time-grounded Question**: Exploring questions based on the time points or intervals of specific entities, such as when certain entities appear or events occur.
- **Spatiotemporal-grounded Question**: Exploring questions based on both the spatial and temporal aspects of specific entities.
- **Entity 2D Spatial Perception**: Inquiring about the 2D spatial position of entities within the video frame, referencing specific areas such as top, bottom, left, right, center, lower-right, lower-left, etc.

Table 7: 3-level Question Categories.

| Level-1 | Level-2 | Level-3 | Example |
|---|---|---|---|
| Perception | Entity Perception | Entity Recognition | Figure 13 |
| | | Entity Counting | Figure 14 |
| | | Entity Attribute | Figure 15 |
| | | Entity State | Figure 16 |
| | Event Perception | Event Recognition | Figure 17 |
| | | Event Counting | Figure 18 |
| | Scene Perception | Scene Recognition | Figure 19 |
| | Text Perception | Text Recognition | Figure 20 |
| | | Text Counting | Figure 21 |
| | Time Perception | Time Localization | Figure 22 |
| | | Time-grounded Question | Figure 23 |
| | | Spatialtemporal-grounded Question | Figure 24 |
| | 2D Spatial Perception | Entity 2D Spatial Perception | Figure 25 |
| Reasoning | Entity Reasoning | Character Identity Reasoning | Figure 26 |
| | | Character Emotion Reasoning | Figure 27 |
| | | Character Intention Reasoning | Figure 28 |
| | | Character Relationship Reasoning | Figure 29 |
| | | Entity General Reasoning | Figure 30 |
| | | Entity Spatial Relationship | Figure 31 |
| | Event Reasoning | Event General Reasoning | Figure 32 |
| | | Event Time Relationship | Figure 33 |
| | | Event Causal Reasoning | Figure 34 |
| | Scene Reasoning | Scene Time Relationship | Figure 35 |
| | Text Reasoning | Text General Reasoning | Figure 36 |
| | | Text Spatial Relationship | Figure 37 |
| | Time Reasoning | Time Interval Reasoning | Figure 38 |
| | | Duration Time Reasoning | Figure 39 |
| Hallucination | Hallucination | Hallucination | Figure 40 41 |

- **Character Identity Reasoning**: Inquiring about the identity of a character, focusing on deducing or identifying who a character is within the context.

- **Character Emotion Reasoning**: Understanding a character's emotions, focusing on analyzing or interpreting the feelings or emotional state of a character.

- **Character Intention Reasoning**: Reasoning about a character's motivations or intentions within the video context, exploring the underlying purpose of actions by analyzing situational details and motivations.

- **Character Relationship Reasoning**: Delving into questions regarding the social or interpersonal relationships between characters, focusing on the type of relationship or connection shared based on observed interactions and context.

- **Entity General Reasoning**: Examining general relationships between entities, including person-object and object-object interactions, clarifying connections or associations beyond spatial or social relationships.

- **Entity Spatial Relationship**: Understanding spatial relationships between objects or entities, focusing on relative positioning to form a mental map of the scene's layout.

- **Event General Reasoning**: Answering questions requiring deeper reasoning or cognitive understanding of events, encouraging a comprehensive interpretation of actions, motivations, and consequences.

- **Event Time Relationship**: Understanding the temporal sequence of events, focusing on ordering events correctly or identifying a particular event's position within a sequence to grasp the flow of actions in the video.

- **Event Causal Reasoning**: Exploring cause-and-effect relationships within events, facilitating an understanding of why an event occurred by linking it to its underlying causes.

- **Scene Time Relationship**: Exploring the sequence in which different scenes occur, focusing on chronological order or progression between different backgrounds.

- **Text General Reasoning**: Answering questions involving inferential content within text in the video, deducing implied meanings, identifying underlying messages, or drawing conclusions from textual information.

- **Text Spatial Relationship**: Addressing the positioning of text within the video frame, such as identifying specific locations (top, bottom, left, right, or center) to clarify the visual layout.
- **Time Interval Reasoning**: Addressing the time interval between two events, focusing on the time gap or separation between occurrences.
- **Duration Time Reasoning**: Exploring the duration of a specific event, inquiring about how long an event lasts or the time span of an action within the context.
- **Hallucination**: Evaluating multiple statements related to video content. Unlike single-statement hallucination questions, the multiple hallucination class involves listing statements and judging which ones are correct. It emphasizes attention to detail and careful assessment of options, distinguishing between accurate and subtly altered statements.

### A.3 QA EXAMPLES

We provide an example for each problem category from Figures 13 to 41.

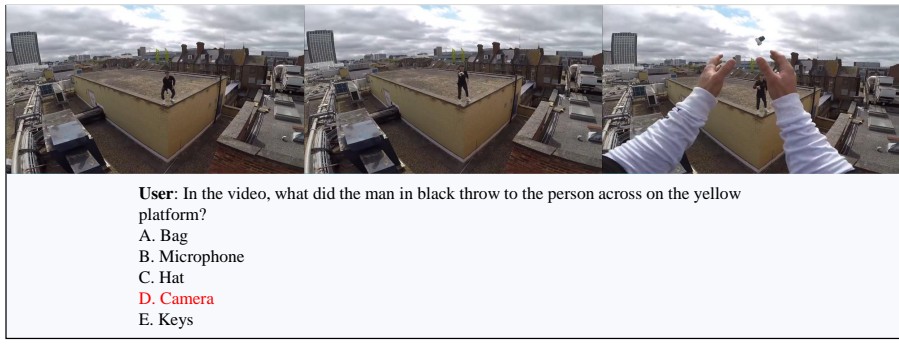

Figure 13: An example of QA in CG-Bench for **Entity Recognition**.

## B MODEL INFERENCE AND EVALUATION

In this section, we list the prompt we use in inference and evaluating existing models.

### B.1 COMMON PROMPTS

**Subtitle Prompt** (Denoted as ):

```
The subtitles of the video are as follows:
<Subtitles>
```

**Subtitle Time Prompt**

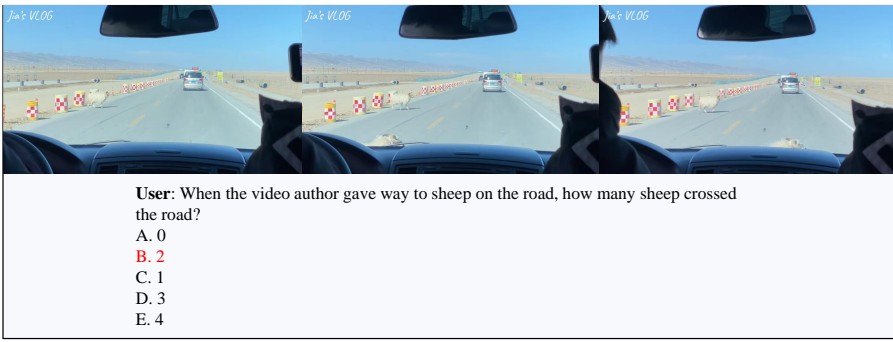

Figure 14: An example of QA in CG-Bench for **Entity Counting**.

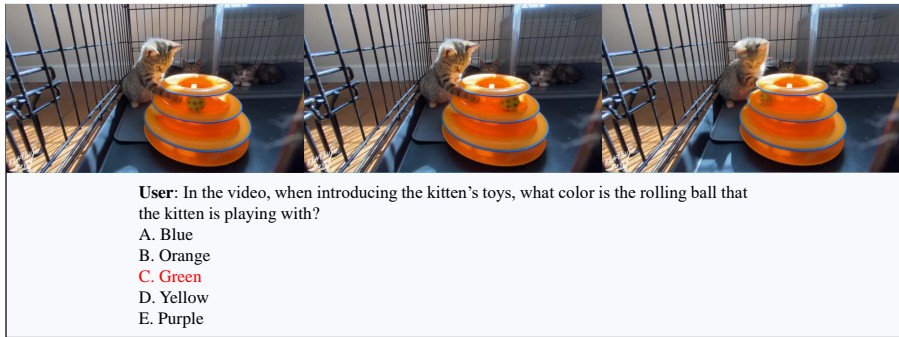

**User**: In the video, when introducing the kitten's toys, what color is the rolling ball that the kitten is playing with?
A. Blue
B. Orange
C. Green
D. Yellow
E. Purple

Figure 15: An example of QA in CG-Bench for **Entity Attribute**.

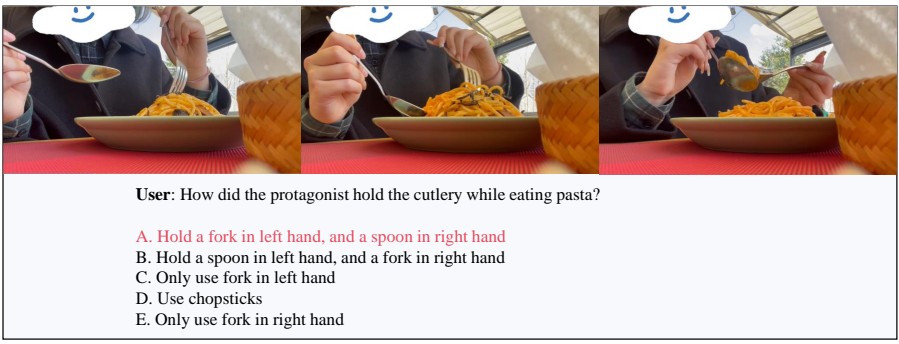

**User**: How did the protagonist hold the cutlery while eating pasta?

A. Hold a fork in left hand, and a spoon in right hand
B. Hold a spoon in left hand, and a fork in right hand
C. Only use fork in left hand
D. Use chopsticks
E. Only use fork in right hand

Figure 16: An example of QA in CG-Bench for **Entity State**.

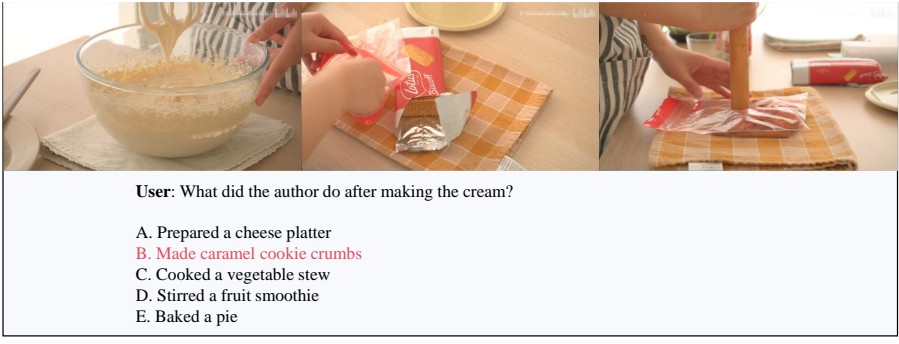

**User**: What did the author do after making the cream?

A. Prepared a cheese platter
B. Made caramel cookie crumbs
C. Cooked a vegetable stew
D. Stirred a fruit smoothie
E. Baked a pie

Figure 17: An example of QA in CG-Bench for **Event Recognition**.

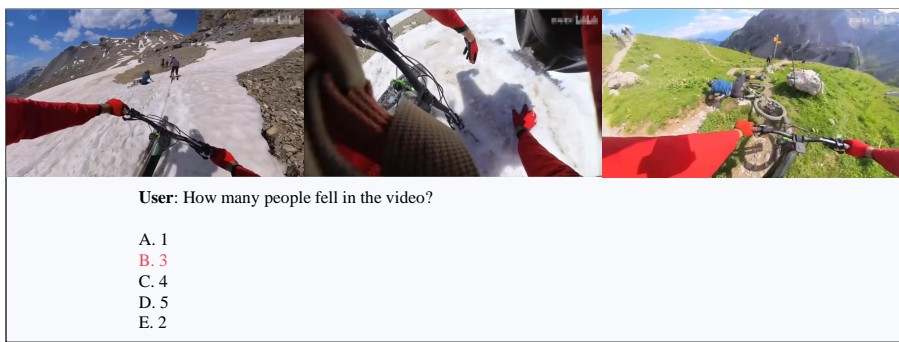

**User**: How many people fell in the video?

A. 1
B. 3
C. 4
D. 5
E. 2

Figure 18: An example of QA in CG-Bench for **Event Counting**.

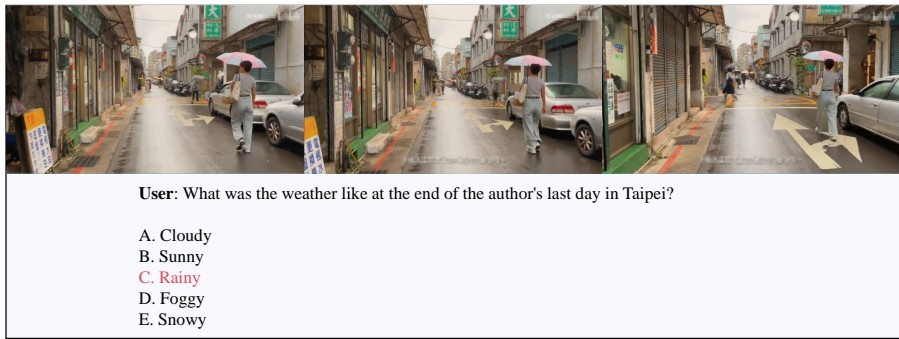

**User**: What was the weather like at the end of the author's last day in Taipei?

A. Cloudy
B. Sunny
C. Rainy
D. Foggy
E. Snowy

Figure 19: An example of QA in CG-Bench for **Scene Recognition**.

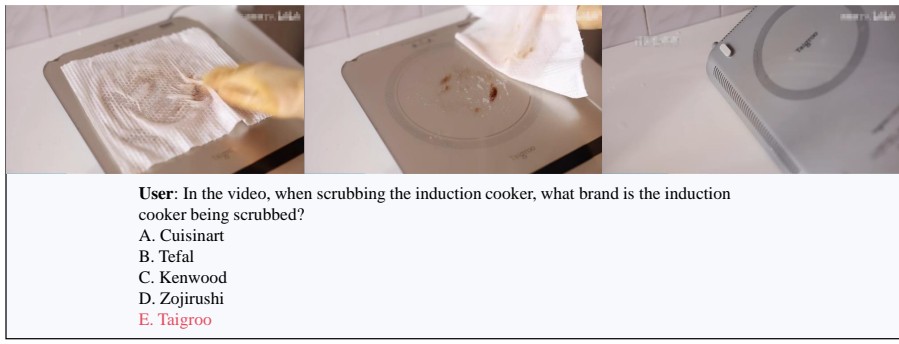

**User**: In the video, when scrubbing the induction cooker, what brand is the induction cooker being scrubbed?
A. Cuisinart
B. Tefal
C. Kenwood
D. Zojirushi
E. Taigroo

Figure 20: An example of QA in CG-Bench for **Text Recognition**.

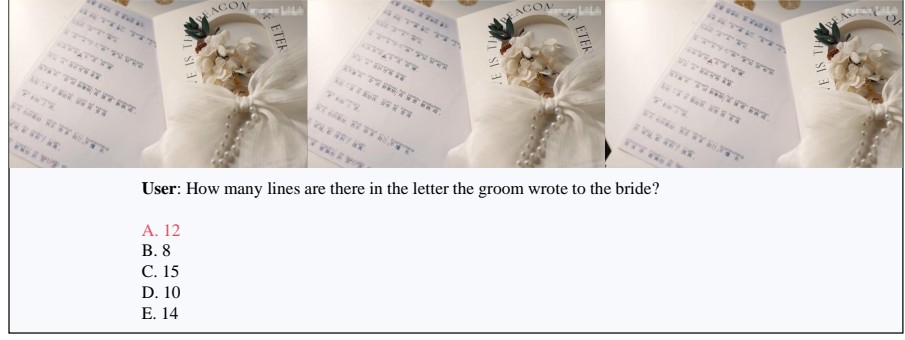

**User**: How many lines are there in the letter the groom wrote to the bride?

A. 12
B. 8
C. 15
D. 10
E. 14

Figure 21: An example of QA in CG-Bench for **Text Counting**.

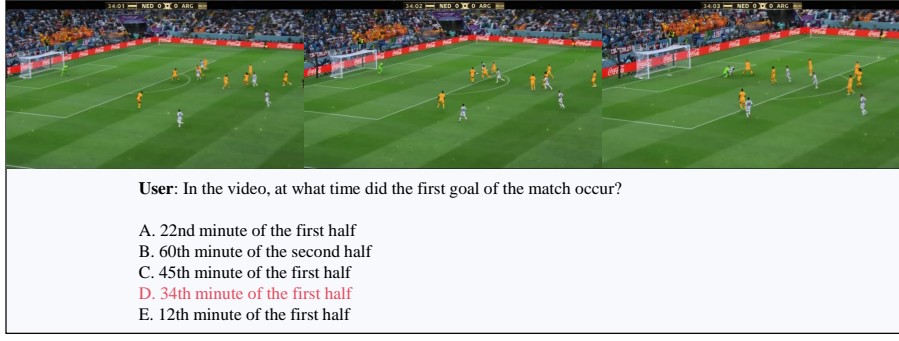

**User**: In the video, at what time did the first goal of the match occur?

A. 22nd minute of the first half
B. 60th minute of the second half
C. 45th minute of the first half
D. 34th minute of the first half
E. 12th minute of the first half

Figure 22: An example of QA in CG-Bench for **Time Localization**.

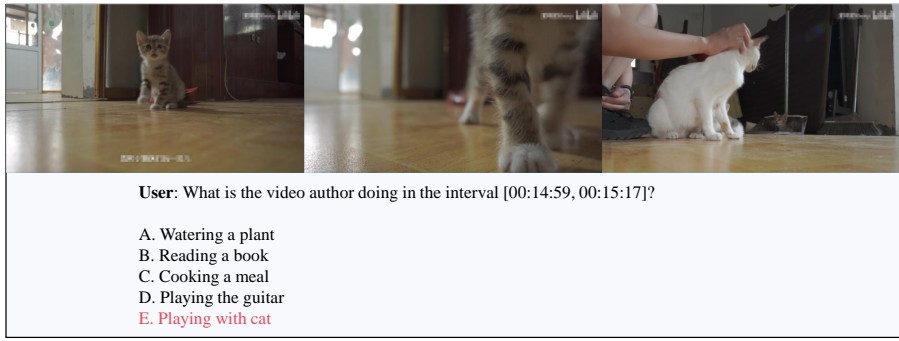

**User**: What is the video author doing in the interval [00:14:59, 00:15:17]?

A. Watering a plant
B. Reading a book
C. Cooking a meal
D. Playing the guitar
E. Playing with cat

Figure 23: An example of QA in CG-Bench for **Time-grounded Question**.

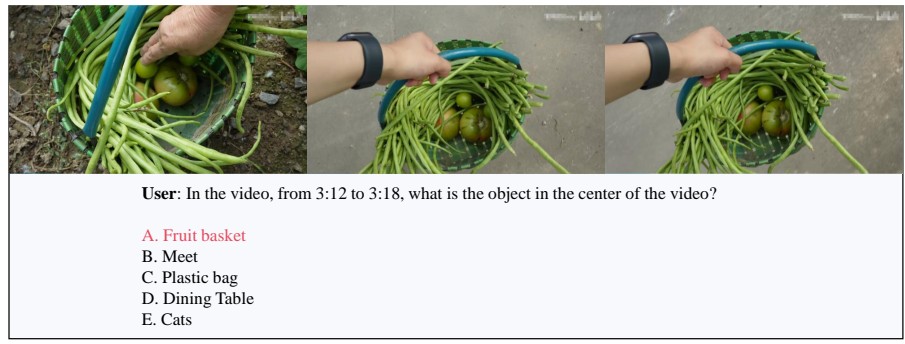

**User**: In the video, from 3:12 to 3:18, what is the object in the center of the video?

A. Fruit basket
B. Meet
C. Plastic bag
D. Dining Table
E. Cats

Figure 24: An example of QA in CG-Bench for **Spatiotemporal-grounded Question**.

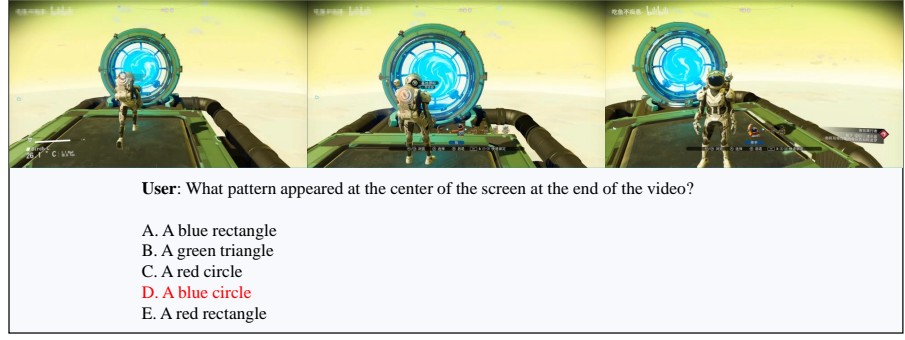

**User**: What pattern appeared at the center of the screen at the end of the video?

A. A blue rectangle
B. A green triangle
C. A red circle
D. A blue circle
E. A red rectangle

Figure 25: An example of QA in CG-Bench for **Entity 2D Spatial Perception**.

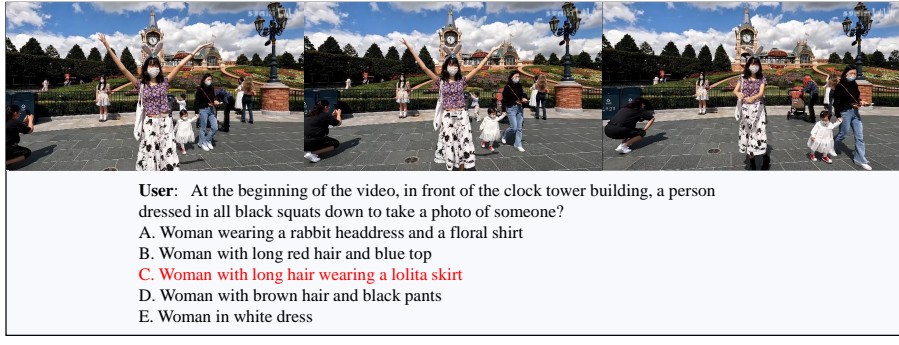

**User**: At the beginning of the video, in front of the clock tower building, a person dressed in all black squats down to take a photo of someone?
A. Woman wearing a rabbit headdress and a floral shirt
B. Woman with long red hair and blue top
C. Woman with long hair wearing a lolita skirt
D. Woman with brown hair and black pants
E. Woman in white dress

Figure 26: An example of QA in CG-Bench for **Character Identity Reasoning**.

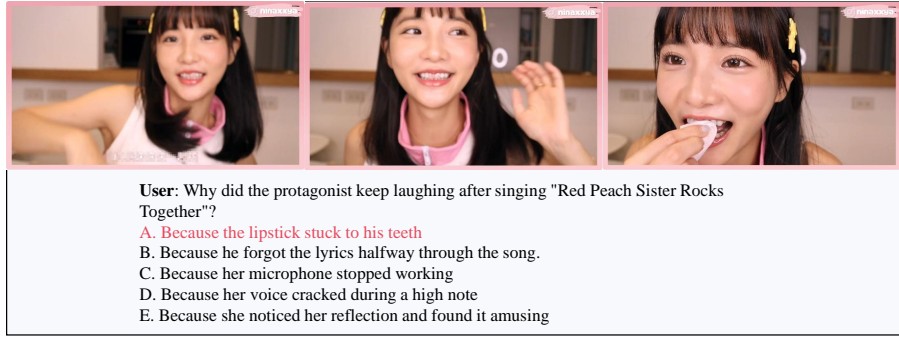

**User**: Why did the protagonist keep laughing after singing "Red Peach Sister Rocks Together"?
A. Because the lipstick stuck to his teeth
B. Because he forgot the lyrics halfway through the song.
C. Because her microphone stopped working
D. Because her voice cracked during a high note
E. Because she noticed her reflection and found it amusing

Figure 27: An example of QA in CG-Bench for **Character Emotion Reasoning**.

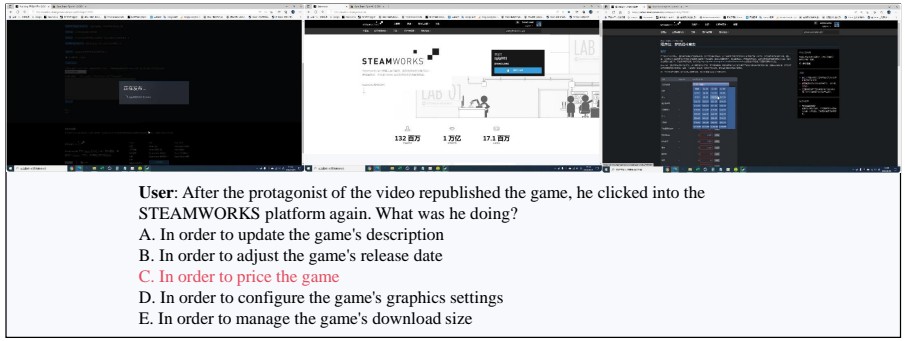

**User**: After the protagonist of the video republished the game, he clicked into the STEAMWORKS platform again. What was he doing?
A. In order to update the game's description
B. In order to adjust the game's release date
C. In order to price the game
D. In order to configure the game's graphics settings
E. In order to manage the game's download size

Figure 28: An example of QA in CG-Bench for **Character Intention Reasoning**.

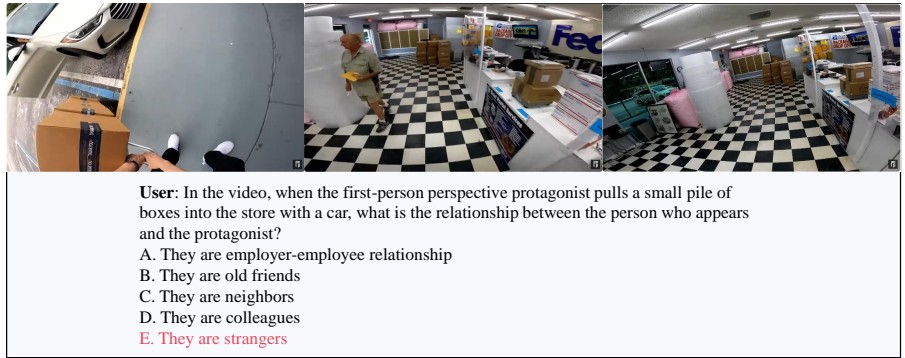

**User**: In the video, when the first-person perspective protagonist pulls a small pile of boxes into the store with a car, what is the relationship between the person who appears and the protagonist?
A. They are employer-employee relationship
B. They are old friends
C. They are neighbors
D. They are colleagues
E. They are strangers

Figure 29: An example of QA in CG-Bench for **Character Relationship Reasoning**.

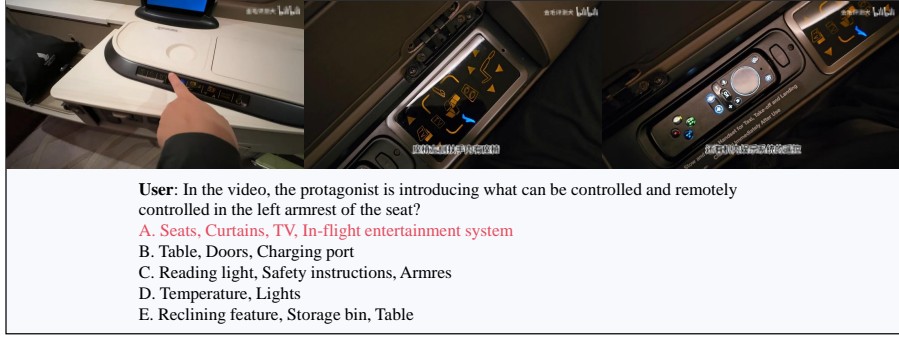

**User**: In the video, the protagonist is introducing what can be controlled and remotely controlled in the left armrest of the seat?
A. Seats, Curtains, TV, In-flight entertainment system
B. Table, Doors, Charging port
C. Reading light, Safety instructions, Armres
D. Temperature, Lights
E. Reclining feature, Storage bin, Table

Figure 30: An example of QA in CG-Bench for **Entity General Reasoning**.

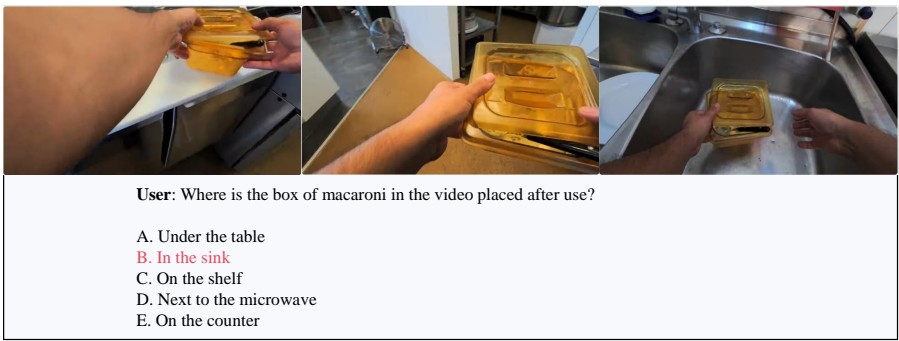

**User**: Where is the box of macaroni in the video placed after use?

A. Under the table
B. In the sink
C. On the shelf
D. Next to the microwave
E. On the counter

Figure 31: An example of QA in CG-Bench for **Entity Spatial Relationship**.

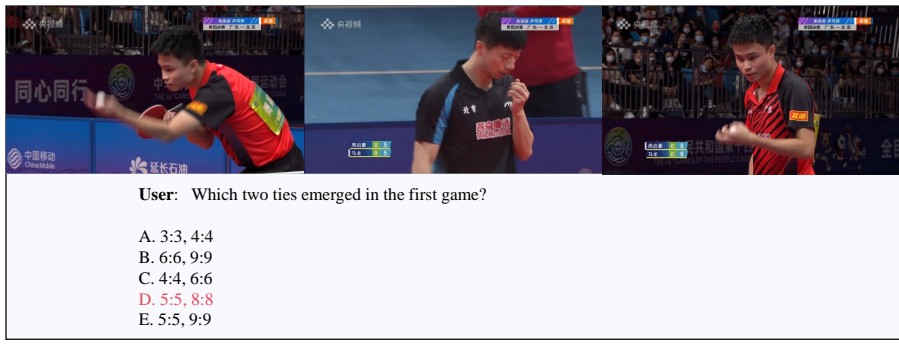

**User**: Which two ties emerged in the first game?

A. 3:3, 4:4
B. 6:6, 9:9
C. 4:4, 6:6
D. 5:5, 8:8
E. 5:5, 9:9

Figure 32: An example of QA in CG-Bench for **Event General Reasoning**.

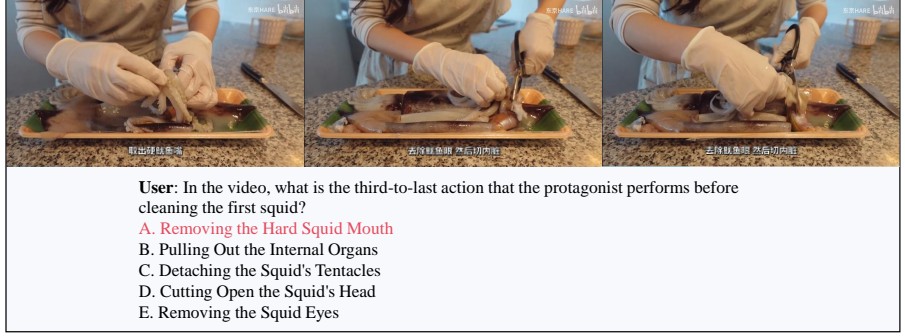

**User**: In the video, what is the third-to-last action that the protagonist performs before cleaning the first squid?

A. Removing the Hard Squid Mouth
B. Pulling Out the Internal Organs
C. Detaching the Squid's Tentacles
D. Cutting Open the Squid's Head
E. Removing the Squid Eyes

Figure 33: An example of QA in CG-Bench for **Event Time Relationship**.

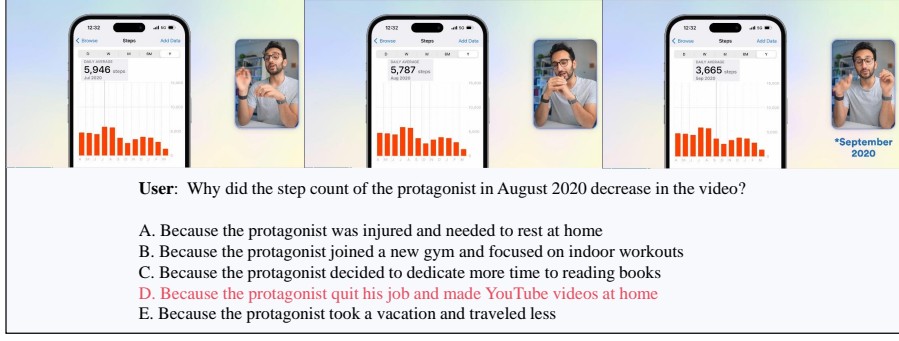

**User**: Why did the step count of the protagonist in August 2020 decrease in the video?

A. Because the protagonist was injured and needed to rest at home
B. Because the protagonist joined a new gym and focused on indoor workouts
C. Because the protagonist decided to dedicate more time to reading books
D. Because the protagonist quit his job and made YouTube videos at home
E. Because the protagonist took a vacation and traveled less

Figure 34: An example of QA in CG-Bench for **Event Causal Reasoning**.

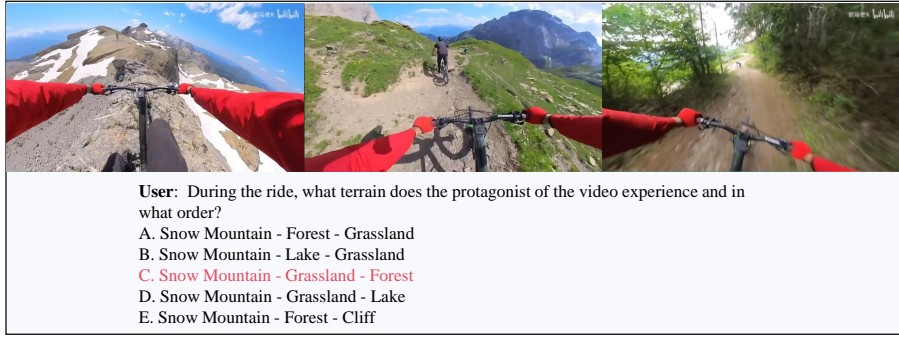

Figure 35: An example of QA in CG-Bench for **Scene Time Relationship**.

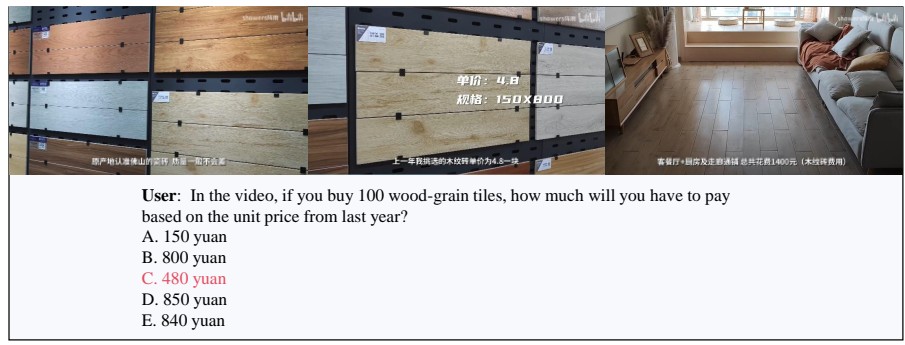

Figure 36: An example of QA in CG-Bench for **Text General Reasoning**.

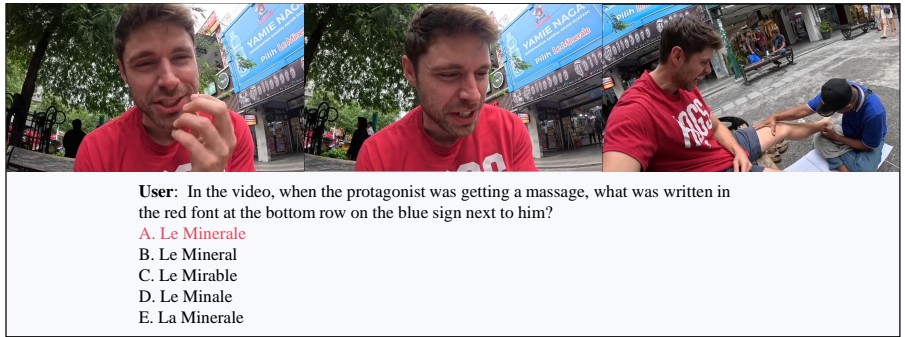

Figure 37: An example of QA in CG-Bench for **Text Spatial Relationship**.

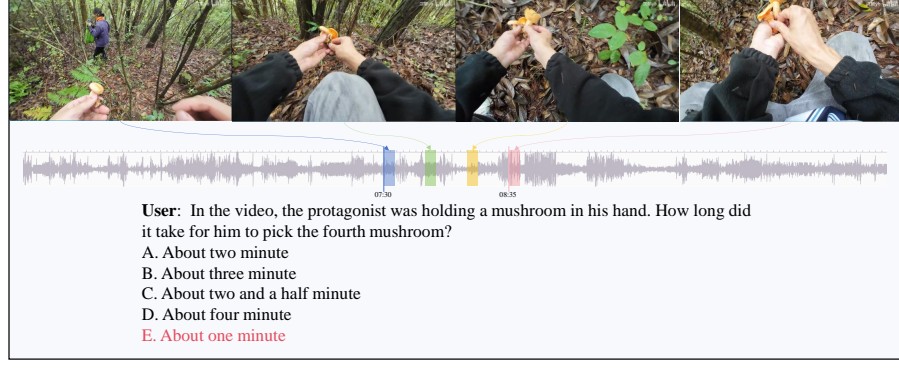

Figure 38: An example of QA in CG-Bench for **Time Interval Reasoning**.

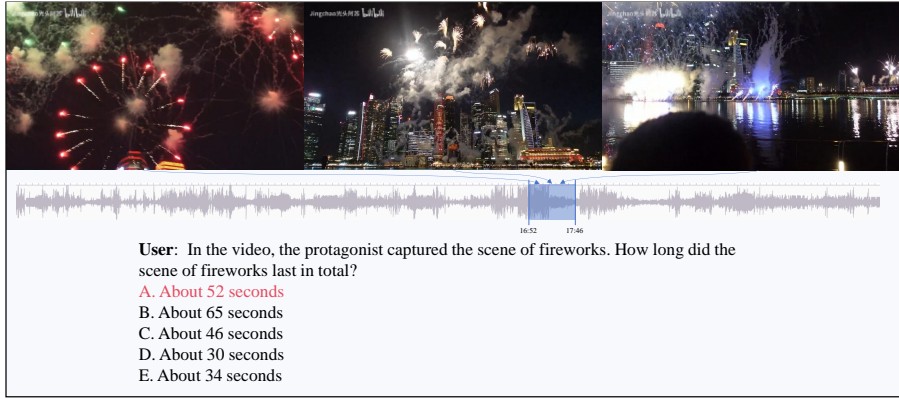

Figure 39: An example of QA in CG-Bench for **Duration Time Reasoning**.

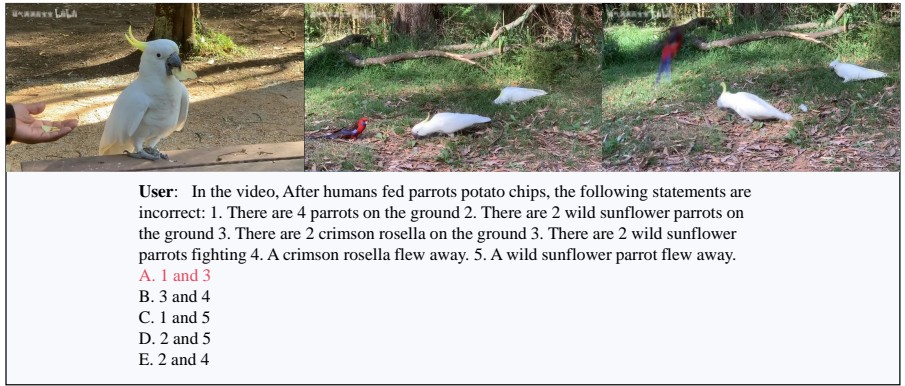

Figure 40: Example 1 in CG-Bench for **Hallucination**.

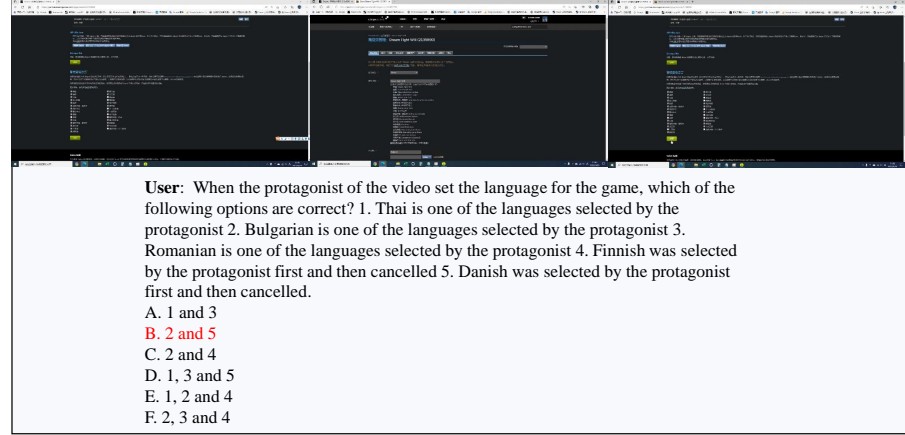

Figure 41: Example 2 in CG-Bench for **Hallucination**.

```
<Subtitle> -> [start, end]: <Subtitle> (Optional)
```

**Frame Time Prompt** (Denoted as <FT>)

```
A total of <n> frames are uniformly sampled from the video, and their
corresponding timestamps are <frame_time1>, <frame_time2>, ...,
<frame_timen>
```

**Choices Prompt** (Denoted as <Choices>)

```
A. ChoiceA
B. ChoiceB
...
E/H. ChoiceE/ChoiceH (5~8 choices)
```

## B.2 INFERENCE PROMPTS

**Long-Video-MCQ & Clue-based-MCQ**

```
Task description:

You will watch a video and read a multiple-choice question based on the
video content. You need to choose an answer that best matches the video
content from five to eight options.

<Frame1>, <Frame2>, ..., <Framen>
 (Optional)
<FT> (Optional)

Multiple-choice question:

<Question>
<Choices>

Important:
- You must only output the uppercase letter corresponding to the
correct answer.
- Do not include any additional text, punctuation, or explanations in
your response.

Your output is:
```

**Blind-MCQ**

```
Task description:

You will be read a multiple-choice question related to a visual task.
However, no visual context or information will be given. Please do your
best to answer the question based solely on the textual information.
Choose the most likely answer from the given options, even if the
question appears to require visual input.

Multiple-choice question:

<Question>
<Choices>

Important:

- You must only output the uppercase letter corresponding to the
correct answer.
- Do not include any additional text, punctuation, or explanations in
your response.
```

```
Your output is:
```

**Question-Clue Grounding**

```
Task description:

You will watch a video and read a multiple-choice question based on the
video content. You need to output each clue interval that can answer
this question in a nested list format.

<Frame1>, <Frame2>, ..., <Framen>
 (Optional)
<FT> (Optional)

Multiple-choice question:

<Question>

<Choices>

Important:

- The output must strictly follow the format: [[start1, end1], [start2,
end2], ...]
where start and end are the timestamps in seconds.
- Any output that does not conform to this nested array format will be
considered incorrect.

Your output is:
```

**Open-Ended QA**

```
Task description:

You will watch a video and read a question based on the video content.
Please answer this question directly based on the frames sampled from
the video.

<Frame1>, <Frame2>, ..., <Framen>
 (Option)
<FT> (Option)

Question:
<Question>

Important:
- You must provide an answer. If explicit clues are lacking, make an
inference. Do your best based on the given frames.
- Failure to provide an inferred answer will be considered incorrect.

Your output is:
```

## B.3 EVALUATION PROMPTS

**Heuristic Evaluation Method for Open-ended QA: Step 1**

```
Task Description:

You are a judge. You will read a question, a model's prediction, and
the ground truth answer to this question. You need to judge whether
the model's prediction is correct. In most cases, this judgment can
be made by determining whether the meaning of the two texts is
consistent. That is, if the meaning of the model's prediction is
consistent with the meaning of the ground truth answer, the prediction
```

```
is considered correct; otherwise, it is considered incorrect. However,
there are some special cases among the incorrect ones, where
inconsistencies may just focus on different details of the same visual
scene and don't have fundamental differences. In this case, the problem
cannot be judged only by text, and additional visual information needs
to be introduced.

Therefore, I hope you:
Output "yes" if the meaning of the two texts of the model's prediction
and the ground truth answer is consistent.
Output "no" if the model's prediction and the ground truth answer are
not consistent, and their meanings are fundamentally different.
Output "need visual clue" if the model's prediction and the ground
truth answer are not consistent but the model's prediction does not
appear to be fundamentally different from the ground truth answer.
It is possible that the two focus on different details of the same
visual scene. Visual information is needed for further judgment.
You are required to give an explanation as to why they might focus
on different details.

Question:
<Question>

The ground truth answer is: "<Answer>"
The model's prediction is: "<Prediction>"

Important:

- The "model's prediction" has already been made based on visual
information. So "need visual clue" means that you need visual
information to make the next judgment, not that the model needs it.
- The "ground truth answer" is annotated by a human, so it is
ABSOLUTELY RIGHT.
Therefore, for relatively simple problems such as counting, if the
model's prediction is different from the ground truth, just output
"no" directly and don't need additional visual information. The only
difference between the "ground truth answer" and the
"model's prediction" that requires further judgment based on visual
information is maybe the different details of the same visual scene
they focus on.

Your output is:
```

**Heuristic Evaluation Method for Open-ended QA: Step 2**

```
Task description:

You are a judge. You will read a question, a model's prediction, and
the sampling frames of the clue intervals of this question. You need
to determine whether the model answered the question correctly based
on the visual information.
I hope you:
- Output "yes", if the model's prediction answers this question
correctly.
- Output "no", if the model's prediction doesn't answer this question
correctly.

Question:
<Question>

The model's prediction is: "<Prediction>"

<Frame1>, <Frame2>, ..., <Framen>
 (Option)
```

```
<FT> (Option)

Your output is:
```

**Pure Text Evaluation Method for Open-ended QA**

```
Task description:

You are a judge. You will read a question, a model's prediction and the
ground truth answer to this question. You need to determine whether the
model answered the question correctly.
I hope you:
- Output "yes", if the model's prediction answers this question
correctly.
- Output "no", if the model's prediction doesn't answer this question
correctly.

Question:
<Question>

The ground truth answer is: "<Answer>"
The model's prediction is: "<Prediction>"
Your output is:
```

**Full Vision-aided Evaluation Method for Open-ended QA: With Ground Truth Answer**

```
Task description:

You are a judge. You will read a question, a model's prediction, the
ground truth answer to this question, and the sampling frames of the
clue intervals of this question. You need to judge whether the model
has answered the question correctly based on the sampling frames of
the clue intervals.

<Frame1>, <Frame2>, ..., <Framen>
 (Option)
<FT> (Option)

Question:
<Question>

The ground truth answer is: "<Answer>"
The model's prediction is: "<Prediction>"
Your output is:
```

**Full Vision-aided Evaluation Method for Open-ended QA: Without Ground Truth Answer**

```
Task description:

You are a judge. You will read a question, a model's prediction, and
the sampling frames of the clue intervals of this question. You need
to judge whether the model has answered the question correctly based
on the sampling frames of the clue intervals.

<Frame1>, <Frame2>, ..., <Framen>
 (Option)
<FT> (Option)

Question:
<Question>

The model's prediction is: "<Prediction>"
Your output is:
```

## C  VIDEO

### C.1  VIDEO COLLECTION

Our videos primarily come from Bilibili and YouTube. We initially constructed broad Level-1 and Level-2 video tags and used these tags for manual searches. During this process, we expanded the Level-2 tags and annotated Level-3 tags. In the manual filtering process, we applied the following criteria:

1. Videos must exhibit sufficient dynamism.
2. For knowledge-related videos, we retained those with some visual dynamism and excluded those that were purely speech-based.
3. We prioritized selecting the most recently uploaded videos to ensure they are as held-out as possible.
4. For each Level-3 tag, we retained only 1-2 videos.
5. We developed a checking program to ensure that the selected video IDs do not overlap with those in major existing video datasets, including COIN (Tang et al., 2019), YouCook2 (Zhou et al., 2018), ActivityNet (Heilbron et al., 2015), HACS (Zhao et al., 2019), CinePile (Rawal et al., 2024), CrossTask (Zhukov et al., 2019), FineGym (Shao et al., 2020a), FineVideo (Farré et al., 2024), HD-VILA-100M (Sun et al., 2022), HiREST (Zala et al., 2023), HowTo100M (Miech et al., 2019), Intern-Vid (Wang et al., 2023), Kinetics (Kay et al., 2017), Mira Data (Ju et al., 2024), OpenVid1M (Nan et al., 2024), Panda70M (Chen et al., 2024d), QueryD (Oncescu et al., 2021), QVHighlight (Lei et al., 2021), Shot2Story (Han et al., 2023), Sports1M (Tran et al., 2019), TAPOS (Shao et al., 2020b), UVO (Wang et al., 2021), VALOR (Chen et al., 2023b), VAST (Chen et al., 2023c), VidChapters (Yang et al., 2024b), VITT (Huang et al., 2020a), Vript (Yang et al., 2024c), YouTubeHL (Sun et al., 2014), YT-Temporal-1B (Zellers et al., 2022), MultiHateClip (Wang et al., 2024a), and ChinaOpen (Chen et al., 2023a).

By this means, approximately 20M video IDs were excluded to ensure that our video data are held out to the largest extent.

### C.2  VIDEO TAGS

We collected 1219 videos on the two platforms, of which 570 videos were collected on YouTube, accounting for 46.8%; and 649 videos were collected on Bilibli, accounting for 53.2%. 50.12% of the videos have subtitles. In addition, we assigned a level-2 or level-3 tag to each video, of which there are 171 level-2 tags and 638 level-3 tags. The specific categories and quantities of tag-2 and tag-3 are shown in Tables 8 and 9.

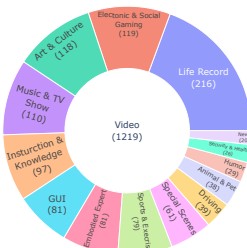

Figure 42: Distribution of video root categories, displaying the number of videos within each category.

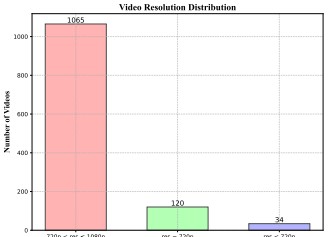

Figure 43: Distribution of video resolusion.

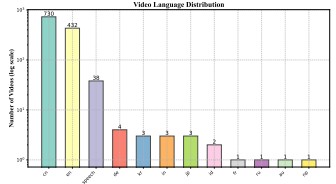

Figure 44: Distribution of video langauge.

#### C.2.1  TAG-1

The categories and quantities of Tag-1 (root categories) are shown in Figure 42.

#### C.2.2  TAG-2

The specific categories and quantities of Tag-2 are shown in Tables 8.

Table 8: Categories and counts of the level-2 video tags.

| Category | # | Category | # | Category | # | Category | # | Category | # |
|---|---|---|---|---|---|---|---|---|---|
| Diverse life | 66 | Beach | 3 | Diet | 47 | Knowledge sharing | 3 | Variety shows | 46 |
| First-person work | 40 | Forest | 3 | Traditional sports | 41 | Board games | 3 | Travel | 34 |
| Extreme sports | 28 | Pet care | 2 | Simulation games | 37 | Russian cuisine | 2 | Movies/TV dramas | 29 |
| Software demonstration | 28 | Racing games | 2 | Wildlife | 24 | MOBA games | 2 | Social games | 25 |
| Festivals | 22 | Driver's license test | 2 | Documentary | 21 | Waterside living | 2 | Play | 24 |
| Coding | 22 | InDesign | 2 | Humor/Comedy | 20 | Designbuilder | 2 | Learning | 21 |
| Working | 18 | Illustrator | 2 | Makeup | 17 | ZBrush | 2 | Eating | 16 |
| Traditional crafts | 16 | Bus | 2 | RPG games | 15 | Digital product reviews | 2 | Shopping sharing | 16 |
| Pets | 14 | Reality challenge games | 2 | Public safety | 13 | Karting | 2 | Fitness | 13 |
| Cooking | 12 | Excavator | 2 | Housekeeping services | 12 | Social news | 2 | Animation | 12 |
| Strategy games | 11 | Helicopter | 2 | Renovation | 11 | Motorcycle | 2 | Handicraft | 10 |
| Funny videos | 10 | Efficiency tool software | 2 | Shopping | 10 | Ruins | 2 | Underwater | 9 |
| Music | 9 | House tour | 2 | Architecture | 9 | Political news | 2 | Humanities | 9 |
| Fashion | 9 | Insects | 2 | Dance | 8 | First-person augmented reality experience | 2 | Technology | 8 |
| Real battlefield/Counter-terrorism | 7 | Business news | 2 | School | 7 | Chemistry | 2 | Open world games | 7 |
| First aid | 6 | Antarctica | 2 | Shooting games | 7 | Debate competition | 2 | In the cave | 7 |
| Medical care | 6 | Human-animal relationship | 2 | Art | 7 | Auction | 2 | Stage performance | 6 |
| Real-time strategy games | 6 | First-person live-action CS | 1 | Board games | 6 | Prison | 2 | Note-taking software | 6 |
| Desert | 6 | Raccoon | 1 | Clothing | 6 | Primates | 2 | Test drive | 5 |
| First-person sports | 5 | Battlefield | 1 | Packing | 5 | Chinese dim sum | 1 | First-person cooking | 5 |
| Aquatic animals | 5 | Installation | 1 | Storage | 5 | Robots | 1 | Cave | 4 |
| Trucks | 4 | Texas Hold'em | 1 | Graphic design software | 5 | Laboratory | 1 | Train | 4 |
| Cars | 4 | Game: Cities Skylines | 1 | First-person driving | 4 | Driver's license | 1 | Space | 4 |
| Knowledge management software | 4 | Photography | 1 | Electric vehicles | 4 | Tea culture | 1 | Comprehensive | 4 |
| Snow | 4 | Comic convention | 1 | Sailing | 4 | Tennis | 1 | Religion | 4 |
| Health and wellness | 3 | First-person adventure | 1 | Airplane | 4 | Motorcycle maintenance | 1 | Repair | 3 |
| Street photography | 3 | Wild | 1 | Selection | 4 | Canyoning | 1 | Beach | 3 |
| Video editing software | 3 | Cycling | 1 | Animation and image generation software | 4 | First-person homework | 1 | Street interviews | 1 |
| Economic news | 1 | First-person work: Coffee shop | 1 | Entertainment news | 4 | Driving | 1 | Diet and wellness | 1 |
| Rescue and disaster relief | 1 | First-person virtual reality experience | 1 | Environmental news | 4 | Sports games | 1 | Music production software | 1 |
| Jade carving | 1 | First-person work: Burger shop | 1 | Detective | 1 | Military news | 1 | Drawing techniques | 1 |
| International news | 1 | Polar animals | 1 | | | First-person games | 1 | | |

### C.2.3 TAG-3

The specific categories and quantities of Tag-3 are shown in Tables 9.

Table 9: Categories and counts of the level-3 video tags.

| Category | # | Category | # | Category | # | Category | # | Category | # |
|---|---|---|---|---|---|---|---|---|---|
| Eight Cuisines | 16 | Photography Tips | 2 | Cat | 5 | Python | 2 | TV Series | 5 |
| Chinese Pastries | 6 | Raft Survival | 2 | Short Film | 5 | Psychology | 2 | Merchandise | 5 |
| Tea Culture | 5 | Portal | 2 | Opera | 5 | Drama | 2 | Giant Panda | 2 |
| Electric Vehicle | 4 | MasterChef | 2 | Pottery | 4 | Food Exploration | 2 | Basketball | 4 |

| Category | # | Category | # | Category | # | Category | # | Category | # |
|---|---|---|---|---|---|---|---|---|---|
| Cleaning Tips | 4 | Action Film | 2 | Football | 4 | MatLab | 2 | Bullet Journal | 4 |
| Sketch | 4 | The Amazing Race | 2 | Motorcycle | 4 | History and Culture: Museum | 2 | Parenting | 3 |
| Grocery Shopping | 3 | Detective Chinatown | 2 | Public Service Short Film | 3 | Space Launch | 2 | Keep Running | 3 |
| Food Delivery | 3 | Unity | 2 | Taiwan Travel | 3 | Prison Documentary | 2 | Dog | 3 |
| Rescue and Disaster Relief | 3 | Kung Fu | 2 | Monopoly | 3 | Golf | 2 | Tennis | 3 |
| Organization Tips | 3 | Pandemic Response | 2 | Grading Homework | 3 | Human-Animal Symbiosis | 2 | Hide and Seek | 3 |
| Extreme Challenge | 3 | The Great British Bake Off | 2 | Dou Dizhu | 3 | The Life We Long For | 2 | Premiere Pro | 3 |
| Comedy | 3 | Shark | 1 | SketchUp | 3 | Puppy | 1 | Stable Diffusion | 3 |
| Meal Prep Tips | 3 | Dumplings | 1 | Winemaking | 3 | Driving Test | 1 | Turkish Cuisine | 3 |
| Photoshop | 3 | Gua Sha | 1 | Economy | 3 | Cardboard | 1 | Japan | 3 |
| Korea Shopping | 3 | VR | 1 | Pr | 3 | Japan Travel | 1 | Divas Hit the Road | 3 |
| Face Painting | 2 | Gourmet Food | 1 | Special Effects Makeup | 2 | Cream Cake | 1 | Everyday Makeup | 2 |
| Campus Life | 2 | Freediving | 1 | Graduation | 2 | Biology/Chemistry Experiments | 1 | Tap Dance | 2 |
| Nursing Procedures | 2 | Biology Experiment | 1 | Escape Room | 2 | Special Forces Training | 1 | Underwater Exploration | 2 |
| Racing | 2 | Surfing | 1 | Rock Climbing | 2 | Horizon | 1 | Wingsuit Flying | 2 |
| Paragliding | 2 | Foundation Makeup | 1 | Gymnastics | 2 | Cake | 1 | DOTA2 | 2 |
| Civilization VI | 2 | Subway | 1 | Plants vs. Zombies | 2 | Pop-up Book | 1 | New Energy Vehicle Test Drive | 2 |
| Novice Highway Driving | 2 | Handmade Soap | 1 | CSGO | 2 | Milk Tea Shop | 1 | GTA5 | 2 |
| Driver's License | 2 | Solo Dining | 1 | Test Drive | 2 | Cheesecake | 1 | Night Market Experience | 2 |
| Housework | 2 | Puff Pastry | 1 | Work Life | 2 | Annual Comedy Competition | 1 | Craft Making | 2 |
| Music MV | 2 | Belly Dance | 1 | Symphony Orchestra | 2 | Trauma Care | 1 | Castle | 2 |
| Underwater Salvage | 2 | Pyramid | 1 | Skiing | 2 | Eyebrow Drawing | 1 | Baseball | 2 |
| Skating | 2 | Parrot | 1 | Counter-Terrorism Action | 2 | Subway Operations | 1 | Rhino | 2 |
| No Man's Sky | 2 | Sushi | 1 | Stardew Valley | 2 | Nail Art | 1 | Supermarket Restocking | 2 |
| Amusement Park | 2 | Meal Prep | 1 | Family Feast | 2 | Underwater Fishing | 1 | Procurement | 2 |
| Magic | 2 | Underwater Welding | 1 | Where Are We Going, Dad? | 2 | Music Festival | 1 | Street Dance of China | 2 |
| Cave | 2 | Rabbit | 1 | Freediving | 2 | Biology | 1 | Cosplay Makeup | 2 |
| Velvet Flowers | 2 | Coffee | 1 | Lantern Festival | 2 | Medicine | 1 | Sailing | 2 |
| Car | 2 | Cultural District | 1 | Truck Driver's Daily Life | 2 | Healthy Living Habits | 1 | Restaurant Waiter | 2 |
| Mountain Village | 2 | Baduanjin | 1 | Trash Picking | 2 | Elephant | 1 | Behind the Scenes | 2 |
| Latin Dance | 2 | Lion | 1 | Medical Equipment Use | 2 | Meerkat | 1 | College Entrance Exam | 2 |
| F1 Racing | 2 | Winter Solstice | 1 | Badminton | 2 | Mediterranean Diet | 1 | Long-Distance Running | 2 |
| Fitness Plan | 2 | Makeup Removal | 1 | Truth or Dare | 2 | Korean Makeup | 1 | Leather Craft | 2 |
| Hanfu | 2 | Shoe Making | 1 | Red Alert 2 | 2 | Freelancer | 1 | Cooking | 2 |
| Shopping | 2 | Mountain Biking | 1 | Theme Park | 2 | Red Panda | 1 | Librarian | 2 |
| Concert | 2 | Brown Bear | 1 | Earthquake Drill | 2 | Wolf | 1 | Snowmobile | 2 |
| Cultural Relics Archaeology | 2 | Oolong Tea | 1 | Embroidery | 2 | Paper Cutting | 1 | Indian Cuisine | 2 |
| Luxury Car Test Drive | 2 | Collage | 1 | Hearthstone | 2 | Vanity | 1 | Vegetarianism | 2 |
| Microfilm | 2 | Mushroom Picking | 1 | Street Dance | 2 | Arab Robe | 1 | Emergency Evacuation | 2 |
| Rescue | 2 | Beading | 1 | Space Station Life | 2 | Beachcombing | 1 | Skateboarding | 2 |
| Diving | 2 | Fishing | 1 | Truck | 2 | Duck House | 1 | Skyline | 2 |
| Ocean Park | 2 | Violin | 1 | Rehabilitation Training | 2 | Dungeon | 1 | Real Battlefield | 2 |
| Water Splashing Festival | 2 | Polar Animals | 1 | Minecraft | 2 | Traditional Chinese Medicine | 1 | Cloud Notes | 2 |
| GoodNotes | 2 | Forza Horizon | 1 | Market Shopping | 2 | Delivery Service | 1 | Antique Market Shopping | 2 |

| Category | # | Category | # | Category | # | Category | # | Category | # |
|---|---|---|---|---|---|---|---|---|---|
| Volleyball | 2 | Convenience Store | 1 | Board Games | 2 | Board Game: Who Are You | 1 | Sculpture | 2 |
| Bus | 2 | Board Game: Storytelling | 1 | Valorant | 2 | Making Small Books | 1 | Notion | 2 |
| City Walk | 2 | Eyebrow Shaping | 1 | Superhero Movies | 2 | Watch Repair | 1 | Train | 2 |
| Fried Chicken | 2 | Concealer | 1 | Zotero | 2 | Laptop | 1 | Duty-Free Shopping | 2 |
| Waterside Life: Beachcombing | 2 | Takoyaki | 1 | CPR | 2 | Creative Market | 1 | Free Fighting | 2 |
| Temple of Heaven | 2 | Variety Show | 1 | National Day | 2 | Board Game: Redemption Journey | 1 | Halloween | 2 |
| Dragon Boat Festival | 2 | Tacit Challenge | 1 | Acupuncture | 2 | Supermarket Challenge | 1 | Ancient Greek Temples | 2 |
| Go-Karting | 2 | Elephants - Wild | 1 | Yacht | 2 | Airplane | 1 | World of Warcraft | 2 |
| After Effects | 2 | Digital Product Review | 1 | Obsidian | 2 | Theme Park | 1 | Pixel Composer | 2 |
| Furniture Assembly | 2 | Digital Product Review: Smart Home | 1 | Digital Painting | 2 | Shopping in Europe | 1 | Digital Product Review: Tablet | 2 |
| Abandoned Buildings | 2 | Digital Product Review: Ergonomic Chair | 1 | Fat Loss Training | 2 | Chocolate Making | 1 | Ab Workout | 2 |
| Hockey | 2 | DIY Mini House | 1 | Spring Festival | 2 | Waterside Life: Fishing | 1 | Easter | 2 |
| Warcraft III | 2 | Digital Product Review: Smartphone | 1 | Wasteland Delivery | 2 | Drawing Techniques | 1 | Pizzeria | 2 |
| High-Altitude Work | 2 | Braised Pork Rice | 1 | Farming | 2 | Fish Pond Construction | 1 | Shopping in Thailand | 2 |
| Museum | 2 | Italy | 1 | Flea Market | 2 | Happy Old Friends | 1 | Art Gallery | 2 |
| Ace vs. Ace | 2 | Wilderness Survival | 1 | I Am a Singer | 2 | Medieval Dynasty | 1 | Firefighting | 2 |
| Military Exercise | 2 | The Witcher | 1 | Snow Survival | 2 | Planet Zoo | 1 | Beach Camping | 2 |
| Dumbbell Training | 2 | Aircraft Loading | 1 | Bowling | 2 | Real-life CS | 1 | Fitness Ball Training | 2 |
| Italian Cuisine | 2 | Car Repair | 1 | Japanese Cuisine | 2 | Pet Store Job | 1 | Elden Ring | 2 |
| Water Obstacle Course | 2 | Ergonomic Chair | 1 | Markdown | 2 | Basement | 1 | Word | 2 |
| CapCut | 2 | Glacier Climbing | 1 | Ruby | 2 | Pufferfish | 1 | VSCode | 2 |
| Blender | 2 | Jade Carving | 1 | Australian Travel | 2 | Ancient Greek Philosophy | 1 | Baking Techniques | 2 |
| Wedding | 2 | Train Driving Simulator | 1 | Drowning | 2 | Theory of Relativity | 1 | Ruins Exploration | 2 |
| Archery | 2 | Used Cars | 1 | Colosseum | 2 | Taiwan Shopping | 1 | Thanksgiving | 2 |
| Autonomous Driving Experience | 2 | AI Painting | 1 | Excavator | 2 | Fishing | 1 | Call of Duty | 2 |
| Adobe Acrobat Pro | 2 | Farm | 1 | Summer Outfits | 2 | Daily Life After Returning Home | 1 | Southeast Asia Travel | 2 |
| Camping | 2 | Home Tour | 1 | Disney | 2 | Village School | 1 | Massage Therapy | 2 |
| The Tonight Show Starring Jimmy Fallon | 2 | Desert | 1 | Fire Drill | 2 | Parkour | 1 | Fire Evacuation | 2 |
| Qipao | 2 | Buddhism | 1 | French Cuisine | 2 | Great Wall | 1 | Helicopter | 2 |
| Manor Lord | 2 | Real-Life Subway Game | 1 | Fallout Shelter | 2 | Mixed Noodles | 1 | Mover | 2 |
| PPT | 2 | Epoxy Resin | 1 | SQL | 2 | Knitting | 1 | Spring Outfits | 2 |
| Seafood Buffet | 2 | Paris | 1 | Studio | 2 | Yoga | 1 | North American Travel | 2 |
| Helicopter Skiing | 2 | Calligraphy | 1 | Qixi Festival | 2 | Thriller | 1 | Spanish Cuisine | 2 |
| German Cuisine | 2 | Real Battlefield/Counter-Terrorism | 1 | inZOI | 2 | Chinese Painting | 1 | Vision Pro | 2 |
| Mailing and Packaging | 2 | Opera | 1 | Making Hot Dogs | 2 | Luggage | 1 | LaTeX | 2 |
| Steam | 2 | Digital Product Review: Electric Toothbrush | 1 | Family Feud | 2 | Mythical Fantasy Film | 1 | Thai Cuisine | 2 |
| Christianity | 2 | Strange House | 1 | Kingdom of Order | 2 | Mahjong | 1 | Plants vs. Zombies Hybrid | 2 |
| Sunny and Warm | 2 | Cat Café | 1 | Grounded | 2 | Kimono (Japan) | 1 | Coffee Shop | 2 |
| JS | 2 | Cleaning | 1 | Quicker | 2 | Editing Tips: Movie Commentary Editing | 1 | Hunting | 2 |
| Department Store Shopping | 2 | Chicago | 1 | Home Gardening | 2 | Market Simulator | 1 | Costume Drama | 2 |

Table 10: Impact of different prompts and modalities on the full test set. Each prompt can be composed of frames (F), frame timestamps (FT), subtitles (S), subtitle timestamps (ST), and audio (A). We conduct the main experiments with GPT4o-0806 (OpenAI, 2024) while studying the audio modality with Gemini-1.5 Pro (Anil et al., 2023).

| model | prompt & modality | clue-acc. | long-acc. | mIoU | Acc@IoU | CRR | OE-acc. |
|---|---|---|---|---|---|---|---|
| GPT4o | S (128 frames) | - | 28.9 | - | - | - | - |
| GPT4o | S (full-video) | - | 31.2 | - | - | - | - |
| GPT4o | F | 66.0 | 52.4 | 3.41 | 10.2 | 79.4 | 35.8 |
| GPT4o | F+FT | 65.1$_{(-0.9)}$ | 52.2$_{(+0.2)}$ | 6.10$_{(+2.69)}$ | 20.6$_{(+10.4)}$ | 80.2$_{(+0.8)}$ | 36.5$_{(+0.7)}$ |
| GPT4o | F+S | 66.1$_{(+0.1)}$ | 53.4$_{(+1.2)}$ | 3.54$_{(+0.13)}$ | 11.0$_{(+0.8)}$ | 80.8$_{(+1.4)}$ | 37.2$_{(+1.4)}$ |
| GPT4o | F+S+ST | 66.3$_{(+0.2)}$ | 52.4$_{(+0.0)}$ | 4.63$_{(+1.22)}$ | 16.3$_{(+6.1)}$ | 78.8$_{(-0.6)}$ | 36.8$_{(+1.0)}$ |
| GPT4o | F+S+FT | 66.5$_{(+0.5)}$ | 52.2$_{(-0.2)}$ | 6.45$_{(+3.04)}$ | 21.3$_{(+11.1)}$ | 78.5$_{(-0.9)}$ | 36.9$_{(+1.1)}$ |
| GPT4o | F+S+ST+FT | **66.5**$_{(+0.5)}$ | **53.9**$_{(+1.5)}$ | **8.33**$_{(+4.92)}$ | **21.7**$_{(+11.5)}$ | **81.1**$_{(+1.9)}$ | **37.2**$_{(+1.3)}$ |
| Gemini | F+S+ST+FT | 61.0 | 43.0 | 7.64 | 18.7 | 70.5 | 18.1 |
| Gemini | F+S+ST+FT+A | 61.2$_{(+0.2)}$ | 43.1$_{(+0.1)}$ | 7.56$_{(-0.08)}$ | 18.6$_{(-0.1)}$ | 70.5$_{(+0.0)}$ | 18.9$_{(+0.8)}$ |

| Category | # | Category | # | Category | # | Category | # | Category | # |
|---|---|---|---|---|---|---|---|---|---|
| Robot Wars | 2 | FamiStudio | 1 | Movie Trailers | 2 | Tattoo Covering | 1 | Snow Mountain Adventure | 2 |
| Equestrian | 2 | Organic Chemistry | 1 | Desert Off-Roading | 2 | Street Food | 1 | Porcelain | 2 |
| Yacht Driving | 2 | Drawing Tips: AI Drawing | 1 | OBS | 2 | Switzerland | 1 | C++ | 2 |
| Clothing | 2 | Iceland | 1 | Dishwashing | 2 | America's Got Talent | 1 | Olympics | 2 |
| Rugby | 2 | New Journey to the West | 1 | Korean Cuisine | 2 | Sand Sculpture Art | 1 | 7 Days to Die | 2 |
| Bartender | 2 | Rafting | 1 | Radiomics | 2 | Battlefield | 1 | European Travel | 2 |
| Livehouse | 2 | Delivery | 1 | Hiking | 2 | Coat | 1 | Ping Pong | 2 |
| Christmas | 2 | Tea Set | 1 | Cat and Mouse Game | 2 | Thailand | 1 | Frostpunk | 2 |
| Black Myth: Wukong | 2 | Interior Design | 1 | First-Person Cooking | 2 | Hengdian | 1 | PC Building | 2 |
| Rainforest Survival | 2 | Who's the Undercover | 1 | High-Intensity Interval Training | 2 | Real-Life Hide and Seek | 1 | The Sinking Land | 2 |

## C.3 VIDEO STATISTICS

We provide an overview of the dataset's characteristics through two statistical visualizations Figure 43 and Figure 44, which demonstrate the distribution of video resolutions, and languages.

Figure 43 illustrates the distribution of video resolutions. The majority of videos (1,065) have a resolution between 720p and 1080p, while 120 videos are exactly 720p. Only 34 videos have a resolution below 720p.

Figure 44 shows the distribution of video languages using a logarithmic scale. The most frequent languages are Chinese (730 videos) and English (432 videos). Additionally, 38 videos have no speech. Other languages such as German, Korean, and Japanese are also represented but in smaller quantities.

## D ADDITIONAL EXPERIMENTS

We further report the ablation studies of different prompts and modalities on the full test subset in Figure 10.

