# OpenReview forum: "CG-Bench: Clue-grounded Question Answering Benchmark for Long Video Understanding"
_ICLR.cc/2025/Conference — ICLR 2025 Poster_

### Official Review · Reviewer_mbFL · 2024-10-27

**Soundness:** 3
**Presentation:** 2
**Contribution:** 3
**Rating:** 6
**Confidence:** 5

**Summary:**

This paper focuses on the evaluation of MLLMs. The authors claimed that existing VideoLLM benchmarks merely focus on either short videos or black evaluation, i.e., simply leveraging MCQ accuracies to measure the model performance without further detailed analysis. Therefore, they proposed CGBench, a novel MLLM benchmark that not only requires the models to predict correct answers for MCQs, but also ground clues through predicting temporal boundaries. The CGBench was created by manually collecting videos and annotated by human annotators, ensuring high quality. Comprehensive evaluations on such a benchmark reveals the limitations of existing MLLMs.

**Strengths:**

1. Overall, the motivation is clear and reasonable: providing a more detailed evaluation for MLLMs rather than simply measuring MCQ accuracies.
2. The created benchmark seems to have good quality, as this was manually annotated by human annotators.
3. The evaluation settings (white box + black box) are reasonable and extensive, jointly provides a good starting point to encourage future research in this direction.

**Weaknesses:**

I have some minor concerns regarding the writing, comparison with existing work, and experiments.

1. The presentation about 'clue' is not very clear. To my understanding, this should be the temporal boundaries (represented by start and end timestamps) of the relative video segments that provide hints to correctly answer the question. If yes, it would be better to state such definition clearly in the main paper.
2. In table 2, a number of existing benchmarks/datasets that also contain the setting of QA + grounding are missing. Further discussion on the differences between CGBench and existing QA + grounding datasets (EgoTimeQA [1], MultiHop-EgoQA [2], ReXTime [3]) and a benchmark (E.T. Bench [4]) is required. And the methods proposed by these papers should also included in table 3, as they are specially designed for such settings.

[1] Grounded Question-Answering in Long Egocentric Videos. CVPR 2024
[2] Grounded Multi-Hop VideoQA in Long-Form Egocentric Videos. arXiv 2024
[3] ReXTime: A Benchmark Suite for Reasoning-Across-Time in Videos. NeurIPS 2024 D&B Track
[4] E.T. Bench: Towards Open-Ended Event-Level Video-Language Understanding. NeurIPS 2024 D&B Track

**Questions:**

Please refer to the weakness part for my concerns and questions. In general, some further discussions shall be conducted.

---

> ### Author Response · Authors · 2024-11-24
>
> **Q1: Definition of Clue.**
>
> Thank you for your feedback. We will clarify the definition of "clue" in the paper. Specifically, "clue" refers to the temporal boundaries (represented by start and end timestamps) of the video segments that provide hints for correctly answering the question. We will include this definition in the relevant section of the paper to enhance clarity.
>
> **Q2: Adding other related benchmarks and models.**
>
> Thank you for your valuable comments. Regarding the missing benchmarks/datasets in Table 2, we have supplemented and discussed them in the revised manuscript. Specifically, we have added detailed discussions on EgoTimeQA, MultiHop-EgoQA, ReXTime, and E.T. Bench, and included the methods proposed by these papers in Table 3.
> 1. In the revised manuscript, we have added detailed discussions on EgoTimeQA, MultiHop-EgoQA, ReXTime, and E.T. Bench, focusing on comparing these datasets with CG-Bench. We highlighted that although these datasets share some similarities with CG-Bench in the QA and grounding settings, their videos are sourced from academic datasets and lack diversity in length, such as Ego4D, QVHighlight, and ActivityNet. In contrast, our CG-Bench primarily focuses on open-domain long videos, thereby introducing more diverse textual queries. Additionally, we have implemented more evaluation mechanisms to provide a more comprehensive assessment.
> 2. In Table 3, we have included the methods proposed by these papers and conducted a detailed performance comparison. We tested GroundVQA, GeLM, and ET-Chat on the Question Grounding and Long-MCQ tasks, with the results as follows:
>
> | model       |  mIoU | Long-Acc |
> |------------|------|----------|
> | GroundVQA   | 3.32 | 27.3     |
> | GeLM         | 5.15 | -        |
> | ET-Chat     | 2.33 | 17.6     |

---

> > ### Comment · Reviewer_mbFL · 2024-11-27
> >
> > Thanks for the response from the authors. I'm holding my original rating.

---

### Official Review · Reviewer_a6oo · 2024-10-29

**Soundness:** 4
**Presentation:** 4
**Contribution:** 3
**Rating:** 8
**Confidence:** 4

**Summary:**

The paper introduces a new benchmark dataset for long video understanding called CG-Bench. In contrast to the few existing long video benchmarks, CG-Bench includes new metrics (white box and black box) that not only measure its ability to answer multiple choice questions, but also the models’ ability to ground its answer in the video by providing time intervals containing clues to the answer. With >12k multiple choice questions, the dataset is the largest of its kind and is fully manually annotated. It includes videos from a wide variety of domains, making it suitable for predicting model performance in many real-world applications. The paper provides an evaluation of >20 models, both open and closed source, across several metrics, providing interesting insights into the state of the art in long video understanding.

**Strengths:**

* There is a large gap between benchmark scores and human performance, showing that there is plenty of room for innovation.
* CG-Bench is the largest long video understanding benchmark to date.
* The ground truth is fully manually annotated.
* Good video lengths: 10-60 mins with mode around 20 mins.
* Good variety of video types and question types.
* The Evaluation of \>20 current methods is a good reflection of the SotA in long video understanding.
* Clue based credibility evaluation is a great contribution to video understanding metrics.
* The paper is clearly structured and well written.

**Weaknesses:**

* The source(s) of the videos are not disclosed and the collection process is not described in enough detail. It would be interesting to know what website(s) the videos come from, what search queries were used. The paper says that videos were manually filtered, but filtering criteria were not mentioned. This information would be critical to understanding the composition and applicability of the benchmark.
* Evaluations on closed-source MLLMs are only performed with up to 32 / 128 frames even though ablations show that performance keeps improving as more frames are used (Fig. 7). Note: Because cost might be a prohibiting factor here, and 128 frames are enough for the provided SotA analysis, I am not considering this shortcoming in my rating. If the benchmark gets publicly released, this could easily be done by groups with sufficient resources.
* Only three examples are provided in the text. It would be good to see a few more example questions from the benchmark to get a feeling for its quality.

**Questions:**

Overall I have no major concerns about this paper. I appreciate the effort the authors put into creating such a large and diverse benchmark and the contributions they are making to video understanding metrics. So I would recommend the paper for publication, but hope the authors could help answer my questions and fix the issues I found in the paper. Also a public release would be much appreciated by the community and increase the paper’s impact.

* Will CG-Bench be publicly released? This would be a major contribution and should be mentioned in the text.  (Though I understand links might need to be omitted due to anonymity reasons.)
* l. 161: Could authors describe what the “Hallucination” question type is and how its questions were designed? Some examples would be helpful too.
* Fig. 5: Why is the clue coverage increasing almost linearly with the time bin index? Is there a reason that clues are more likely to be located towards the end of the video than the beginning?
* l. 176 “To minimize expression loss, annotators use their native language during the annotation process.“ Are questions translated into English? If yes, how? If not, how many languages does the benchmark contain and what is their distribution?
* Do the annotators have access to audio and/or subtitles of the video? If yes, were any measures taken to make sure the answers are grounded in visual content and not only in spoken / audio content?
* l. 298: If \\tau is 0, doesn’t acc.@IoU degenerate to MCQ-acc? Is this a typo?
* l. 305: “In other words, the accuracy of Long-Video MCQ (long-acc.) should be greater than or equal to the accuracy of Clue-based MCQ (clue-acc.).” This contradicts Tab.3, where clue-based accuracy is always greater than long video accuracy. Also the formula for CCC assumes that long video accuracy is smaller than clue based accuracy. My guess is that this is due to context dilution, i.e. the model not being able to extract relevant information when more irrelevant information is added to the context. So, CRR is actually a measure of robustness to context dilution and not a measure of the model’s ability to seek out clues outside the provided clue windows, as the “Black box evaluation” paragraph says. I would suggest rephrasing this paragraph accordingly.
* l. 323: “To address this, we leverage a low-hallucination MLLM, such as GPT-4o” I’m curious why GPT-4o is called a low-hallucination MLLM. Is there any literature to back this up?
* Tab. 3: Why are mIoU scores \> 1? According to Eq. 1, they should be \<=1.
* Tab. 4:
  * I was surprised that accuracy does not vary much as modalities are added. clue-acc changes from 66.0 to 66.5 and long-acc from 52.4 to 53.9 as subtitles and timestamps are added. I would assume that subtitles would give models a clear advantage since they are a strong semantic signal. Why do they have so little effect?
  * It would be interesting to have an ablation that uses only subtitles and no frames.
* Tab. 5:
  * What is the unit for bias? What does a score of 1.0 mean?
  * How are trigger rate and recall rate defined? This explanation could be expanded in the text.
* Fig. 7d: GPT-4o’s CRR decreases at first and stays below other models until it catches up at 128 frames. I’m curious of authors have an explanation for this.
* l. 458: What is the meaning of “refusal rate” here?
* l. 464: “When evaluators were provided only with ground truth (GT) and visual information, the bias between human scores and model-based evaluations increased.“ Does this mean when only one of the modalities is provided?
* l. 465: “While fully leveraging visual information improved evaluation accuracy \[...\]” What does “fully leveraging visual information” refer to here? Does this refer to the “GT+Vis” columns?

**Minor points:**

* l. 107: “Despite the continuous proposal of various MLLMs, their real-world performance in long video understanding is still unknown.” This statement is not entirely fair since other long video benchmarks such as Video-MME have provided evaluations of these models. I would suggest toning down the statement a little.
* l. 192: “we align with annotators in real-time“ I am not sure I understand what this means.
* l. 214: What are “option probabilities”?
* In related work, it would help if all method/dataset names were mentioned alongside citations, e.g. Llava-Next (Zhang et al., 2024b).
* In tables, please right-align numerical columns for easier comparison.
* l. 232, use \\citep
* l. 372: Tab. 3 has many more open source models. Maybe add “among others”.
* Tab. 4: In row 2, long-acc column, the sign of the diff should be minus. In the following row, the diff is 1.0.

---

> ### Author Response · Authors · 2024-11-24
>
> **Q1: Process of collecting and filtering videos.**
> Thank you for your valuable feedback. Our videos primarily come from Bilibili and YouTube. We initially constructed broad Level 1 and Level 2 video tags and used these tags for manual searches. During this process, we expanded the Level 2 tags and annotated Level 3 tags. In the manual filtering process, we applied the following criteria:
> 1. Videos must exhibit sufficient dynamism.
> 2. For knowledge-related videos, we retained those with some visual dynamism and excluded those that were purely speech-based.
> 3. We prioritized selecting the most recently uploaded videos to ensure they are as held-out as possible.
> 4. For each Level 3 tag, we retained only 1-2 videos.
> 5. We developed a checking program to ensure that the selected video IDs do not overlap with those in major existing video datasets, including coin, youcook2, activityNet, hacs, cinepile, Crosstask, finegym, finevideo, hdvila100m, hirest, howto100m, internvid, kinetics, Mira data, openvid1m, panda70m, queryd, qvhighlight, shot2story, sports1m, tapos, uvo, valor, vast, vidchapters, vitt, vript, youtubehl and yt-temporal-1b, MultiHateClip and ChinaOpen.
>
> We will include these details in the revised version to enhance the understanding of the benchmark's composition and applicability.
>
> **Q2: Open-source benchmark data.**
>
> We will release the data within a few days. To avoid the issue of video download failure, we will provide a solution that can definitely download the video which will not bring license issues.
>
> **Q3: more examples and Hallusion question.**
>
> We have added more QA and QAC examples in the supplementary materials. Here is one example of Hallusion Question.
> ```
> Question: When the protagonist of the video set the language for the game, which of the following options are correct?
> 1. Thai is one of the languages selected by the protagonist
> 2. Bulgarian is one of the languages selected by the protagonist
> 3. Romanian is one of the languages selected by the protagonist
> 4. Finnish was selected by the protagonist first and then cancelled
> 5. Danish was selected by the protagonist first and then cancelled
> Option:
> A. 1 and 3;
> B. 2 and 5;
> C. 2 and 4;
> D. 1, 3 and 5;
> E. 1, 2 and 4;
> F. 2, 3 and 5
> ```
> Different from the general binary discrimination hallucination test questions, we adopted a multiple-choice setting to find multiple correct answers or multiple wrong answers from multiple statements. This significantly increases the difficulty and efficiency of the hallucination test.
>
> **Q4: Figure of clue coverage.**
>
> Thank you for your insightful comment. We discovered that the increase in clue coverage was due to a missing normalization step in our code when generating the plot. We have corrected this error and updated the figure in the revised version of the paper.
>
> **Q5: Translation of questions.**
>
> Yes, all questions are translated into English. Initially, we experimented with GPT-4 for translation, but we observed unstable results. Therefore, we ultimately used Google Translate for this task. After translation, we conducted a manual review of the QA pairs to ensure accuracy and maintain the integrity of the original expressions.
>
> **Q6: How to make sure the answers are grounded in visual content?**
>
> We require annotators to annotate questions that contain clues in the video, mainly using vision as the anchor. A small number of samples can use audio, audio + video, or subtitles as anchors to enhance the multimodal evaluation capabilities of the benchmark. During the Review Iteration process, we manually check whether most annotated questions are queries about vision.
>
> **Q7: Acc@IoU with τ=0.**
>
> Regarding acc@IoU when τ=0, it does not simply reduce to accuracy. Simple accuracy does not account for the model's ability to identify the clue interval. In contrast, acc@IoU with τ=0 requires the model to not only select the correct option but also produce a time interval that overlaps at least slightly (tIoU > 0, not >= 0) with the annotated clue interval.
>
> **Q8: Rephrasing the definition of CRR.**
>
> Thank you for the valuable suggestion. We have rephrased the description about CRR as the reviewer suggested. Please refer to the revised manuscript for this change.
>
> **Q9: Why GPT4o is a low hallucination model ?**
>
> Thank you for your comment. Low hallucination is a relative concept. We refer to GPT-4o as a low-hallucination MLLM because of its outstanding performance on several well-known benchmarks, such as OpenCompass and the lmsys leaderboard. In the revised manuscript, we will add relative descriptions to convey this more clearly.
>
> **Q10: why are mIoU scores >1 ?**
>
> Thank you for your reminder. This value is in percentage and we have modified it in the revised manuscript.

---

> > ### Comment · Reviewer_a6oo · 2024-11-26
> >
> > Thank you for the detailed responses to my questions! Your responses have been quite insightful, so I would suggest adding them to the manuscript as well if you have not already done so. Overall, the responses have alleviated my concerns. I'm glad to hear this benchmark will be publicly released and I'm sure it will be very valuable for the video understanding community.

---

> ### Author Response · Authors · 2024-11-24
>
> **Q11: Abalation studies of Subtitles.**
>
> We conducted an ablation study focusing on subtitles, with the results as follows (f-128 indicates using subtitles corresponding to uniformly sampled 128-frame timestamps, and full-video indicates using all subtitles from the full video):
>
> | prompts                  | long-acc  |
> |-----------------------|-------|
> | subtitle (f-128)      | 28.9  |
> | subtitle (full video) | 31.2  |
>
> We have updated these results in the revised manuscript. The experimental results indicate that although subtitles provide useful semantic signals, their benefit is significantly diminished when visual input is used. This also indirectly demonstrates that our benchmark is more inclined towards visual signals.
>
> **Q12: More explaination of Table 5.**
>
> In table 5, Bias is the absolute value of the difference between the human evaluation score and the model evaluation score (in percent). Trigger rate refers to the probability that the model triggers "visual clues required". Recall rate refers to the recall rate of the model triggering "visual clues required" given the manually annotated trigger annotations.
>
> **Q13: Curve of CRR for GPT4o.**
>
> The short-acc of each model is a model-related constant, leading to a strong correlation between the CRR curve and the long-acc curve. Both GPT-4o and Gemini exhibit a curve that initially declines before rising, which I believe is influenced by their respective training strategies. As commercial models, GPT-4o and Gemini possess extensive conversational capabilities, resulting in less stable performance on multiple-choice questions (MCQs), especially in sparse contexts. In contrast, Qwen2VL appears to be more benchmark-oriented, possibly undergoing comprehensive training for MCQs, enabling it to "guess" answers more effectively from visual information.
>
> **Q14: What is the "refusal rate" of Gemini.**
>
> When using Gemini, it sometimes rejects questions for some reason. There are two main reasons: 1. Gemini thinks the question involves privacy issues. 2. Gemini thinks there is not enough information to answer the question. According to our statistics, Gemini's rejection rate is about 31%.
>
> **Q15: Experiments of open-ended evaluation quality (L464-L466).**
>
> Thank you for comments. Your understanding is correct, we have added the explanation in the revised paper.

---

### Official Review · Reviewer_eKZm · 2024-11-03

**Soundness:** 3
**Presentation:** 3
**Contribution:** 3
**Rating:** 6
**Confidence:** 4

**Summary:**

This paper propose a new video language question-answering evaluation benchmark, featuring that each QA pair is grounded into a time interval. The QA and the time interval (as clues for the question) are manually annotated by human raters which are useful to answer the question. Having the correct time interval prediction ensures the MLLM finds the correct evidence to answer the question. The authors propose a various set of evaluation metrics, including overall QA accuracy, QA accuracy given ground truth time intervals, time intervals prediction IoUs, and GPT-guided open-ended evaluation. The authors evaluated a wide range of existing MLLMs and observed performance gaps between the models.

**Strengths:**

- The problem of evaluating multi-modal large language model is important, and I believe this paper provided a unique perspective of evaluating the temporal grounding for video question answering.

- The authors provided a comprehensive set of evaluation metrics which all make sense to me. The clue-acc, long-acc, and CRR metrics smartly shows how the model leverage temporal information without explicitly look at temporal grounding prediction. I also like how the authors use GPT-4o as a guide with optional visual inputs to save evaluation budgets.

- The authors evaluated a wide range of existing MLLMs (Table 3) and ablated several interesting settings (Table 4 and Figure 7), with interesting insights (e.g., how the models take use of more frames, how additional modality helps).

- The fact that Audio does not help much of the dataset is a good surprise to me, where I was under the impression that many existing long video datasets are heavily ASR-biased.

- I like the fact that the authors collect videos and questions all from scratch rather than reusing existing datasets or create by using LLM. This is important to reduce biases in existing datasets.

**Weaknesses:**

- The mIoU scores in Table 3 for most models are low (<9%), and even the human performance is only 35.5%. This raises a question that how reliable are these temporal interval annotations are. At a high level, a model does not need to look at all frames within the interval to gather the necessary information, but only a few frame or a much smaller interval is sufficient. Therefore metrics like recall may help here (this is optional for the rebuttal).

- Following the above comment, it is unclear to me how good the current MLLMs are at following the specific output format instruction in supplementary Line 552 "Question-Clue Grounding". It will be helpful if the authors report how often the case "Any output that does not conform to this nested array format will be considered incorrect" happens, to get a better understanding of why mIoUs are low.

- Since the authors collect the videos manually, please make sure there are no license issues of releasing the videos.

**Questions:**

Overall this paper puts good effort in an important direction, with solid dataset designs and experiments. I have some concerns on the mIoU evaluation as discussed in paper weakness, and expect the authors to reply them. My current rating is a week accept, and I am happy to further raise the ratting in my concerns are addressed in the rebuttal.

---

> ### Author Response · Authors · 2024-11-24
>
> **Q1: Reliability of interval annotation and Recall@IoU**
>
> Regarding the reliability of temporal interval annotations, this issue has been widely discussed in early temporal understanding works such as Charades-STA[1] and ActivityNet-Caption[2]. Due to the high-level semantic nature of events, there is often ambiguity in event boundaries, leading to low agreement among annotators. This is why we provide human performance on mIoU to relax the optimal target and make the evaluation more realistic.
> We agree that using Recall@IoU is another valuable metric to consider. We have conducted preliminary calculations, and the results are as follows:
>
> | threshold   | 0.1   | 0.2   | 0.3   | 0.4   | 0.5   | avg |
> |-------|-------|-------|-------|-------|-------|--------|
> |   Recall@IoU  | 24.29 | 15.82 | 10.67 | 6.47  | 4.06  |  12.262      |
>
> However, we believe that mIoU still holds reference value, and we plan to retain both metrics. The related content has been updated in the manuscript. We appreciate the reviewer for the valuable suggestion.
>
>
> **Q2: Impact of prompts on outputing interval formats.**
>
> This sentence, "Any output that does not conform to this nested array format will be considered incorrect.", serves as a "warning" to the models in the prompt. Before using this sentence, the success rates for GPT-4o and Gemini in following the nested array format instructions were approximately 62.3% and 53.5%, respectively. After adding this sentence, the success rates improved to 91.0% and 82.4%. In actual testing, outputs that did not conform to the required format were not considered incorrect; instead, we requested the output again until the model produced the correct format. Therefore, all final predictions could be used to compute valid tIoUs.
>
> **Q3: License issues for releasing videos.**
>
> To avoid the issue of video download failure, we will provide a solution that can definitely download the video which will not bring license issues.
>
> **Reference**
>
> [1] Gao J, Sun C, Yang Z, et al. Tall: Temporal activity localization via language query.
>
> [2] Iashin V, Rahtu E. Multi-modal dense video captioning.

---

### Official Review · Reviewer_6Ngt · 2024-11-04

**Soundness:** 2
**Presentation:** 3
**Contribution:** 3
**Rating:** 6
**Confidence:** 3

**Summary:**

The paper introduces CG-Bench, a benchmark for assessing multimodal large language models on long video understanding, moving beyond traditional multiple-choice questions (MCQs). CG-Bench emphasizes “clue-grounded” question answering by focusing on models' ability to retrieve relevant video segments that support accurate question answering. The dataset includes 1219 long videos annotated with over 12K Q&A pairs spanning categories like perception, reasoning, and hallucination. CG-Bench not only support standard evaluation metrics such as accuracy, but also incorporates two novel metrics for credibility to measure the reliability of model predictions: (1) white-box evaluation, where models need to identify specific intervals related to questions, and (2) black-box evaluation, which measures the discrepancy in model performance when with access to the full video compared to solely relying on the clue interval.

**Strengths:**

- Originality:  The paper presents CG-Bench, a new benchmark specifically designed for clue-grounded question answering in long video understanding, which fills a notable gap in multimodal large language model (MLLM) evaluation on video understanding. In addition, the paper also proposed two novel metrics for measure reliability of model predictions specifically to the clue-grounded QA scenario in CG-Bench.
- Quality: The quality of this work is reflected in the construction of the dataset, with 1,219 long videos, over 12,000 Q&A pairs, and a detailed three-tier tagging system, ensuring comprehensive coverage across categories like perception, reasoning, and hallucination.
- Clarity: The paper is easy to follow.
- Significance: The proposed benchmark has great potential for MLLM research, as it exposes the limitations of current models in long video comprehension and sets a new standard for assessing MLLMs in real-world, temporally extended contexts.

**Weaknesses:**

The paper can be improved by incorporating more details regarding the construction of the benchmark, and more thorough evaluation of existing models to provide more insights on how and why existing models fail to solve CG-Bench, to inspire future works. Please refer to the Questions section for detailed concerns.

**Questions:**

- What is the distribution of the resolution of the raw videos used in this dataset?
- Can the authors provide more details about the video collection process? How to prevent the videos from overlapping with the pre-training data of existing MLLMs, especially if they are collected from the internet? The reviewer would imagine this as a difficult process especially with closed source models.
- As subtitles are provided with the videos, are all subtitles human uploaded or machine transcribed? Are all subtitles in English?
- The reviewer is also curious about how the performance of both closed-source and open-source MLLMs can be affected by (1) the frame sampling strategy (e.g., uniform vs. key frames) and (2) the frame resolution (as it also affects how many frames can be fit in memory for open-source MLLMs given the computational limit).
- For Black-Box Evaluation, the authors claim that "a model with access to the full video should yield higher accuracy compared to solely relying on the clue interval.". The reviewer find it a little counter-intuitive, though the full video may contain more information, however, it is a harder task (even for human) to search for the clue first then answer the question, than answer the question with clue provided.
- How to think of the white-box credibility measures especially when the meta information such as timestamps is not available to the model?  For example, evaluation setting with only frames as inputs (F in Table 4), the accuracy is on par or slightly higher than the setting with both frames and frame time prompt (F+FT in Table 4), while the scores for white-box evaluation are significantly lower.
- Regarding the quality control process, especially for the "small model & sparse frame check" step, can the authors provide more insights in terms of what kind of questions are being filtered?

---

> ### Author Response · Authors · 2024-11-24
>
> **Q1: Distribution of video resolution.**
>
> We have analyzed the distribution of video resolutions. Our findings show that there are 1,065 videos with a short side between 720 and 1080 pixels, 120 videos with a short side exactly 720 pixels, and 34 videos with a short side less than 720 pixels. This indicates that the majority of our videos are high-definition.
>
> **Q2: How to collect videos and ensure the videos are held out?**
>
> Thank you for your valuable feedback. Our videos primarily come from YouTube and BiliBili. We initially constructed broad Level 1 and Level 2 video tags and used these tags for manual searches. During this process, we expanded the Level 2 tags and annotated Level 3 tags. In the manual filtering process, we applied the following criteria:
> 1. Videos must exhibit sufficient dynamism.
> 2. For knowledge-related videos, we retained those with some visual dynamism and excluded those that were purely speech-based.
> 3. We prioritized selecting the most recently uploaded videos to ensure they are as held-out as possible.
> 4. For each Level-3 tag, we retained only 1-2 videos.
> 5. We developed a checking program to ensure that the selected video IDs do not overlap with those in major existing video datasets, including coin, youcook2, activityNet, hacs, cinepile, Crosstask, finegym, finevideo, hdvila100m, hirest, howto100m, internvid, kinetics, Mira data, openvid1m, panda70m, queryd, qvhighlight, shot2story, sports1m, tapos, uvo, valor, vast, vidchapters, vitt, vript, youtubehl, yt-temporal-1b, MultiHateClip and ChinaOpen.
> We will include these details in the revised version to enhance the understanding of the benchmark's composition and applicability.
>
> **Q3: Where are the subtitles from?**
>
> Our subtitles are downloaded directly from the video website. However, not all videos have subtitles. According to our statistics, 50.12% of videos have subtitles, mainly in English and Chinese, and a small amount of other languages.
>
> **Q4: Effect of frame sampling strategies.**
>
> We conducted relevant experiments. In order to speed up the testing process, we mainly tested GPT4o at 50 frames. The experiment was divided into three parts: low resolution, high resolution, and key frame extraction (by ffmpeg) + low resolution. The results are as follows:
> | resolution & frame sampling               | long-acc   |
> |--------------------|--------|
> | Low-res, uniform   | 46.7   |
> | Low-res, keyframe  | 45.71  |
> | High-res, uniform  | 50.96  |
>
> The results show that higher resolution can bring some improvement, but keyframes have no significant effect. We have updated the table in the revised paper.
>
> **Q5: More explanation on Black-box evaluation**
>
> In general, providing a model with more information should enhance the accuracy of question answering. Ideally, the accuracy with the full video (long-acc) should be greater than or at least equal to the accuracy with only the clue interval (clue-acc). This is primarily because the human-annotated clue intervals may not always represent the optimal clue segments; for instance, the actual clues might extend beyond the annotated intervals. Content outside the clue interval may also indirectly and implicitly help in the reasoning process. A sufficiently sampled and sufficiently trained long-video MLLM should be capable of automatically identifying all useful relevant clues.
> Therefore, under the same sampling settings, CRR can be used to assess the extent to which the same model has been effectively trained on long contexts. For cross-model comparisons, CRR can evaluate how is alignment extent of different models between both short and long contexts. For example, with the same 32/128 sampling settings, the CRR for Gemini is 70.5, while for GPT-4o it is 81.1. This indicates that Gemini's performance in matching short and long contexts is weaker compared to GPT-4o.
>
> **Q6: Explanation on white-box evaluation without timestamp information.**
>
> In general, most existing MLLMs treat videos as a series of images. When multiple images are provided without the corresponding timestamps, the model lacks the temporal context necessary to process these images as a cohesive video. This absence makes it challenging for the model to perform comprehensive temporal dynamic modeling, leading to some uncertainty in the model's output and resulting in suboptimal performance. Adding additional temporal information helps improve tasks such as temporal localization.

---

> ### Author Response · Authors · 2024-11-24
>
> **Q7: QA examples filtered by the "small model & sparse frame check" step.**
>
> Regarding the quality control process, especially for the "small model & sparse frame check" step, we appreciate the reviewer’s request for more insights.
> In general, questions lacking temporal dynamics are easily addressed by sparse frame checks, while small models excel at solving the corresponding simpler questions. For instance:
> ```
> Question 1: What color is the protagonist's cycling outfit in the video?
> A. Red
> B. Blue
> C. Green
> D. Black
> E. White
> ```
> > Analysis: The protagonist is cycling in a first-person view, and the outfit appears throughout the video.
>
> ```
> Question 2: What does the climbing wall in the video look like?
> A. An indoor artificial climbing wall
> B. A climbing wall with various color markers
> C. An outdoor low-altitude climbing wall
> D. A training wall with multiple climbing routes
> E. A cliff face
> ```
> > Analysis: The climbing wall is a prominent target, and the distinctions between the options are very clear, requiring minimal comprehension.
>
> These examples illustrate how the "small model & sparse frame check" step effectively filters questions that do not require extensive temporal understanding or complex reasoning.

---

> ### Comment · Reviewer_6Ngt · 2024-11-27
>
> Thanks the authors for the detailed response. The reviewer still have a few doubts, detailed below:
> - If subtitles are not available for 50% of the videos, are Table 4 results on the whole dataset, or just the subset that have subtitles? If on the whole dataset, Table 4 results do not correctly reflect the impact of adding subtitle or subtitle timestamps.
> - Black-box evaluation: If possible, can the authors provide examples where (1) the human-annotated clue intervals may not always represent the optimal clue segments and (2) content outside the clue interval may also indirectly and implicitly help in the reasoning process.
> - White-box evaluation without timestamp information: the reviewer's confusion is mainly on how the models would respond for clue grounding when no timestamp information is provided to the model. Especially looking at the prompt in L1518, the models are required to respond in seconds. When only the frames are provided, it is nearly impossible for even humans to ground the clues in seconds. Hence, the reviewer thinks mIoU in this case is not very meaningful when timestamps are not provided.

---

> ### Author Response · Authors · 2024-11-29
>
> **Q1: Ablation studies on the subset with subtitles.**
>
> Our original intention was to establish a connection between Table 3 and Table 4 by presenting the performance metrics for the complete dataset. This dataset includes videos without subtitles; however, their inclusion does not affect the comparisons made in the ablation experiments.
>
> | Model   | Prompt & Modality | Clue-Acc. | Long-Acc. | mIoU | Acc@IoU | CRR  | OE-Acc. |
> |---------|-------------------|-----------|-----------|------|---------|------|---------|
> | GPT4o   | S (128 frames)    | -         | 31.5      | -    | -       | -    | -       |
> | GPT4o   | S (full-video)    | -         | 34.3      | -    | -       | -    | -       |
> | GPT4o   | F                 | 65.8      | 51.8      | 3.39 | 10.7    | 78.7 | 35.4    |
> | GPT4o   | F+FT              | 65.3      | 51.6      | 5.73 | 20.4    | 79.0 | 36.8    |
> | GPT4o   | F+S               | 66.7      | 53.4      | 3.96 | 11.2    | 80.1 | 38.2    |
> | GPT4o   | F+S+ST            | 67.1      | 54.1      | 5.19 | 13.2    | 80.6 | 38.4    |
> | GPT4o   | F+S+FT            | 67.4      | 53.2      | 7.80 | 22.3    | 78.9 | 37.9    |
> | GPT4o   | F+S+ST+FT         | 67.5      | 54.9      | 9.68 | 26.7    | 81.3 | 39.5    |
> | Gemini  | F+S+ST+FT         | 62.1      | 45.1      | 9.16 | 20.7    | 72.6 | 23.2    |
> | Gemini  | F+S+ST+FT+A       | 62.3      | 45.0      | 9.10 | 19.8    | 72.2 | 23.5    |
>
> After your valuable feedback, we realized that including the entire dataset obscures the true impact of different prompts and modalities on performance because the gain values are averaged. To address this, we recalculated the performance metrics using a subset of videos that include subtitles only, as detailed below:
> In the revised manuscript, we will replace Table 4 with this new table and move the original Table 4 to the appendix as an additional reference.
>
> **Q2: Examples for black-box evaluation**
>
> Generally speaking, the two situations you mentioned cannot be completely decoupled. To demonstrate this situation comprehensively, we found an example from our dataset. We also provide the corresponding online video link for your reference.
>
>
> Example:
>
> video_link: https://www.youtube.com/watch?v=WHAMiF-YJtQ
>
> Question: What did the two protagonists do after dinner in Florence, Italy?
>
> Choices: A. Watch the sunset over city B. Exchange gifts C. Hug and kiss D. Enjoy a man singing E. Drink together
>
> Direct clue: [21:30, 21:45] This segment shows the two protagonists hugging and kissing after dinner in Florence. It is directly relevant to the question.
>
> Indirect clues:
> 1. [[7:41, 7:50], [9:49, 9:54], [23:12, 23:16]] These intervals show moments where the protagonists share intimate moments, like kissing each other, drinking the same bottle of water, and being close to each other. These moments can imply a closer relationship, helping reason that after dinner, they would engage in physical affection (hugging and kissing).
>
> 2. [[11:12, 11:35], [12:51, 13:30], [16:17, 16:28], [20:15, 21:00]] These intervals show iconic landmarks and activities in cities like Venice and Rome. This is useful to help filter out irrelevant parts of the video, ensuring the answer stays focused on Florence. It also helps exclude other possible and similar scenarios in different cities.
>
> Suboptimal clue: [21:36, 21:45]. This segment only shows the two protagonists hugging and kissing. However, the question asks "After dinner in Florence". The "dinner in Florence" part should also be included in the clue annotation.
>
>
> **Q3: how the models would respond for clue grounding when no timestamp information is provided to the model?**
>
> In the case where the testing without the timestamps (row F/F+S in table-4), we prompt the model that these images are sampled uniformly from the entire video and require it to output the normalized time interval between 0 and 1. During the evaluation, we convert the normalized output into an interval of seconds to calculate the tIoU.

---

> > ### Author Response · Authors · 2024-12-03
> >
> > Dear Reviewer 6Ngt, as the discussion deadline is approaching, we would like to check if there are any other questions. We believe our additional responses have adequately addressed your doubts. If there are any further questions or concerns, we are happy to provide more information.
> >
> > Thank you for your efforts in helping us to improve the submission!

---

### Official Review · Reviewer_3P2q · 2024-11-04

**Soundness:** 4
**Presentation:** 3
**Contribution:** 3
**Rating:** 5
**Confidence:** 4

**Summary:**

This paper presents a newly collected dataset that is intended for evaluating long-video (10min+) understanding capability of current models. The key is to introduce clue-grounded questions as question-answer-clue triplet, to minimize the possibility of models "cheating" by combining short-video understanding and elimination to "answer correctly" without genuinely understand long videos. The dataset contains 1219 manually selected videos and 12K QAC triplets with fine-grained category annotations. The authors also designed two clue-based metrics as complementatry to the multiple-choice accuracy metric, to better assess whether model understands the video before they can answer correctly. The authors evaluated the proposed dataset with multiple open-sourced and commercial LLMs and showed that existing LLMs still have much room for improvement on this novel dataset.

**Strengths:**

* Originality

For long video understanding, the lack of appropriate benchmark dataset has been a main problem for properly evaluating modeling progress, and actually many concurrent work is aiming for filling this gap. This paper presents a novel dataset with high quality (human annotation) and novel setup (clue-guided), essentially a finer-grained segment-level annotation, which enabled more possibility such as the finer-grained evaluation (the two clue-based evaluation metrics). The idea is not entirely new but the dataset can be highly impactful for the community.

* Quality

The dataset is curated in mutiple rounds with a lot of human efforts, and the authors have done extensive experiments with modern MLLM models as well as human evaluations to prove its usefulness.

* Clarity

The paper is well-written and is rather easy to follow.

* Significance

The proposed benchmark dataset could be a useful addition to the community.

**Weaknesses:**

* In L151 the authors mention that videos are manually collected from Internet to "avoid using videos that have been used for pre-training", but no details is provided. Actually I'm curious how this can be guaranteed. Is it done by license? The human-annotated QAC might be novel but I feel it's hard to ensure that Internet videos are excluded from LLM pre-training.

* I fully understand the difficulty (e.g. high requirement for hardware) to evaluate long videos especially some long-tailed videos (longer than 60min), but using 16/32 or even 128 frames probably is far from enough. One possible mitigation is to perform human eval under the same constraints, e.g. only feeding human with 16/32 frames and ask them to answer the question. This probably can give a more informative sense about how challenging the dataset could be. Also it would be great to have finer-grained analysis, e.g. grouping video/questions based on the duration (10min, 10-20min, etc.) and show how severe the "undersampling" issue is for longer video. Actually this could be a quite important missing ablation.

* The two proposed metrics (white-box and black-box evaluations) are intuitive but really simple; it feels more like some analysis to provide insights but not as novel, e.g. the white-box evaluation is clearly an extension from video temporal grounding. Also even though the authors claim the acc@IoU to be a combined metric, but since $\tau=0$ it's effectively degraded to MCQ accuracy.

**Questions:**

* See my comments about weakness for questions.

---

> ### Author Response · Authors · 2024-11-24
>
> **Q1: How to ensure the videos are held out?**
>
> This is indeed a good question. It is challenging to ensure with absolute certainty that the video data has not been used in model pre-training, especially for commercial models that may have utilized all the available online content, including validation and test sets of academic datasets. For our video selection, we focused on hindering overlapping with publicly available academic datasets. To achieve this, we implemented two main strategies:
> 1. We selected videos from the most recently uploaded content.
> 2. In addition, we developed a filtering tool and excluded all the videos that have already existed in major video datasets. This is done by checking whether their video IDs already exist in these datasets, including coin[1], youcook2[2], activityNet[3], hacs[4], cinepile[5], Crosstask[6], finegym[7], finevideo[8], hdvila100m[9], hirest[10], howto100m[11], internvid[12], kinetics[13], Mira data[14], openvid1m[15], panda70m[16], queryd[17], qvhighlight[18], shot2story[19], sports1m[20], tapos[21], uvo[22], valor[23], vast[24], vidchapters[25], vitt[26], vript[27], youtubehl[28], yt-temporal-1b[29], MultiHateClip[30] and ChinaOpen[31].
>
> By this means, approximately 20M video IDs were excluded to ensure that our video data are held out to the largest extent.
>
> **Q2: Human performance on sparse uniformly sampling.**
>
> Thank you for your insightful comments.
> To address your suggestion, we conducted a quick human evaluation experiment under constrained visual conditions. We uniformly sampled 30 videos from CG-Bench, resulting in 296 questions. For each video, we uniformly sampled 128 frames and asked volunteers to perform a multiple-choice question (MCQ) testing. The resulting accuracy was 59.85%.
> This result indicates that our dataset is indeed challenging and that it is difficult to derive solutions from a limited number of frames.
> *It also highlights that even the most advanced models, such as GPT-4o, have ample room for improvement in long video comprehension.*
>
> **Q3: Performance grouped by video duration.**
>
> Thank you for your insightful suggestion regarding a finer-grained analysis. We have conducted the analysis by grouping videos based on their duration and assessing the long-acc performance of GPT-4o with 128 frames. The results are as follows:
>
> | video duration group         | long-acc  |
> |------------------|-------|
> | 10-20 minutes    | 55.6  |
> | 20-30 minutes    | 53.4  |
> | 30-40 minutes    | 54.0  |
> | 40-50 minutes    | 53.6  |
> | 50-60 minutes    | 50.7  |
> | 60+ minutes      | 49.4  |
>
> These results indicate that the current model is facing undersampling issues, particularly with longer videos.
> We will release the code for this grouped analysis alongside our open-source materials to facilitate further exploration and validation by the research community.
>
> **Q4: Metric design and Acc@IoU with τ=0.**
>
> We acknowledge that the accuracy and tIoU metrics are not novel; however, they are widely accepted and reliable within the community. Our focus is on how combining these metrics can provide more insightful evaluations. Regarding acc@IoU when τ=0, it does not simply reduce to accuracy. Simple accuracy does not account for the model's ability to identify the clue interval. In contrast, acc@IoU with τ=0 requires the model to not only select the correct option but also produce a time interval that overlaps at least slightly (tIoU > 0, not >= 0) with the annotated clue interval. We have updated the corresponding part in the main manuscript for a clearer explanation.

---

> ### Author Response · Authors · 2024-11-24
>
> **Reference**
>
> [1] Tang Y, Ding D, Rao Y, et al. Coin: A large-scale dataset for comprehensive instructional video analysis.
>
> [2] Zhou L, Xu C, Corso J. Towards automatic learning of procedures from web instructional videos.
>
> [3] Caba Heilbron F, Escorcia V, Ghanem B, et al. Activitynet: A large-scale video benchmark for human activity understanding.
>
> [4] Zhao H, Torralba A, Torresani L, et al. Hacs: Human action clips and segments dataset for recognition and temporal localization.
>
> [5] Rawal R, Saifullah K, Basri R, et al. Cinepile: A long video question answering dataset and benchmark.
>
> [6] Zhukov D, Alayrac J B, Cinbis R G, et al. Cross-task weakly supervised learning from instructional videos.
>
> [7] Shao D, Zhao Y, Dai B, et al. Finegym: A hierarchical video dataset for fine-grained action understanding.
>
> [8] Farré, Miquel and Marafioti, et al. FineVideo.
>
> [9] Sun Y, Xue H, Song R, et al. Long-form video-language pre-training with multimodal temporal contrastive learning.
>
> [10] Zala A, Cho J, Kottur S, et al. Hierarchical video-moment retrieval and step-captioning.
>
> [11] Miech A, Zhukov D, Alayrac J B, et al. Howto100m: Learning a text-video embedding by watching hundred million narrated video clips.
>
> [12] Wang Y, He Y, Li Y, et al. Internvid: A large-scale video-text dataset for multimodal understanding and generation.
>
> [13] Carreira J, Zisserman A. Quo vadis, action recognition? a new model and the kinetics dataset.
>
> [14] Ju X, Gao Y, Zhang Z, et al. Miradata: A large-scale video dataset with long durations and structured captions.
>
> [15] Nan K, Xie R, Zhou P, et al. Openvid-1m: A large-scale high-quality dataset for text-to-video generation.
>
> [16] Chen T S, Siarohin A, Menapace W, et al. Panda-70m: Captioning 70m videos with multiple cross-modality teachers.
>
> [17] Oncescu A M, Henriques J F, Liu Y, et al. Queryd: A video dataset with high-quality text and audio narrations.
>
> [18] Lei J, Berg T L, Bansal M. Detecting moments and highlights in videos via natural language queries.
>
> [19] Han M, Yang L, Chang X, et al. Shot2story20k: A new benchmark for comprehensive understanding of multi-shot videos.
>
> [20] Karpathy A, Toderici G, Shetty S, et al. Large-scale video classification with convolutional neural networks.
>
> [21] Shao D, Zhao Y, Dai B, et al. Intra-and inter-action understanding via temporal action parsing.
>
> [22] Wang W, Feiszli M, Wang H, et al. Unidentified video objects: A benchmark for dense, open-world segmentation.
>
> [23] Chen S, He X, Guo L, et al. Valor: Vision-audio-language omni-perception pretraining model and dataset.
>
> [24] Chen S, Li H, Wang Q, et al. Vast: A vision-audio-subtitle-text omni-modality foundation model and dataset.
>
> [25] Yang A, Nagrani A, Laptev I, et al. Vidchapters-7m: Video chapters at scale.
>
> [26] Huang G, Pang B, Zhu Z, et al. Multimodal pretraining for dense video captioning.
>
> [27] Yang D, Huang S, Lu C, et al. Vript: A Video Is Worth Thousands of Words.
>
> [28] Sun M, Farhadi A, Seitz S. Ranking domain-specific highlights by analyzing edited videos.
>
> [29] Zellers R, Lu J, Lu X, et al. Merlot reserve: Neural script knowledge through vision and language and sound.
>
> [30] Wang H, Yang T R, Naseem U, et al. Multihateclip: A multilingual benchmark dataset for hateful video detection on youtube and bilibili.
>
> [31] Chen A, Wang Z, Dong C, et al. Chinaopen: A dataset for open-world multimodal learning.

---

> ### Comment · Reviewer_3P2q · 2024-11-26
>
> Thanks for the explanation and additional experiments! Sorry that my initial review might be not clear enough regarding $\tau=0$ -- a naive dummy solution under such constraint is basically predicting full span for all time so it would ensure overlap>0 always holds. Any comments on that?

---

> > ### Author Response · Authors · 2024-11-27
> >
> > We agree with the reviewer's perspective on this evaluation metric. In fact, we have previously considered this issue: the model only needs to output the complete video to cover the necessary clues.
> >
> > In our initial experiments, we used thresholds like acc@0.1, acc@0.2, acc@0.3, acc@0.4, and acc@0.5, which are standard metrics for temporal localization tasks. The results were as follows (value between 0 and 1):
> >
> > ```
> > threshold: 0.1, acc@iou: 0.14935064935064934
> > threshold: 0.2, acc@iou: 0.09925788497217068
> > threshold: 0.3, acc@iou: 0.0686456400742115
> > threshold: 0.4, acc@iou: 0.04452690166975881
> > threshold: 0.5, acc@iou: 0.029684601113172542
> > acc@iou avg 0.06524427952999381
> > ```
> >
> > However, we found that even for GPT-4o, the values were quite low, potentially narrowing the comparative differences between models due to the task's difficulty. We assumed the current model did not intentionally optimize for this metric but would generate responses based on its actual capabilities. As a result, we relaxed the evaluation criteria to τ=0 in our original submission.
> > In light of your valuable feedback, it is indeed possible for model developers to hack this metric, which is detrimental to advancing long-form video understanding.
> > Therefore, we will adjust this metric and use a higher threshold for a more robust evaluation.
> > Specifically, we set five thresholds 0.1, 0.2, 0.3, 0.4, 0.5 to calculate respective Acc@IoU, and then take their average as the final result.
> > We will revise the relevant sections in the paper, and we welcome further suggestions for improvement.

---

> > > ### Author Response · Authors · 2024-12-03
> > >
> > > Dear Reviewer 3P2q, as the discussion deadline is approaching, we would like to check if there are any other questions.
> > > We strongly believe our responses have adequately addressed your concerns. Should there be any further questions, we are happy to provide more information.
> > >
> > > Thank you for your efforts in helping us to improve the submission!

---

### Author Response · Authors · 2024-11-24

We sincerely thank all reviewers for their valuable feedback. To address the reviewer's concern, we have provided comprehensive responses to each concern in a point-by-point manner, including new experimental results. To make improvements based on the suggestions, we have updated the manuscript accordingly, revising the corresponding parts and adding more experiments in the appendix.

---

### Meta-Review · Area_Chair_84J8 · 2024-12-22

**Metareview:**

This paper proposes CG-Bench, a benchmark for evaluating multimodal LLMs on long video understanding, featuring >1K videos and over 12,000 question-answer-clue triplets focused on clue-grounded question answering. It introduces metrics for assessing segment retrieval accuracy and performance differences with and without clue segments, revealing significant room for improvement in existing models. The paper is well written and is a valuable contribution to the Video LLM community by providing a challenging benchmark in the long video understanding domain. The clue-guided setup is novel and has human annotations. The weaknesses of the paper are its very low human baseline (35.5%), raising questions about the quality of the annotations and the solvability of the task and the unclear setting of video licenses. Overall the paper still provides a clear and positive contribution for the community, both empirically and methodologically and the AC recommends acceptance.

**Additional Comments On Reviewer Discussion:**

There was good discussions between the reviewers and the authors which led to clarifications on the presentation and methodology, the impact on prompts on outputting interval formats and the refusal rate of Gemini MLLM. While the point about the unclear video downloadability/licence remains (Videos come from YouTube and Bilibili), the authors maintain that a clear and stable solution will be available. Conditional on this, the AC recommends acceptance.

---

### Decision · Program_Chairs · 2025-01-22

Accept (Poster)